# Coincident glutamatergic depolarizations enhance GABA$_A$ receptor-dependent Cl$^-$ influx in mature and suppress Cl$^-$ efflux in immature neurons

**Aniello Lombardi**[1], **Peter Jedlicka**[2,3,4], **Heiko J. Luhmann**[1], **Werner Kilb**[1]*

**1** Institute of Physiology, University Medical Center Mainz, Johannes Gutenberg University, Mainz, Germany, **2** ICAR3R - Interdisciplinary Centre for 3Rs in Animal Research, Faculty of Medicine, Justus-Liebig-University, Giessen, Germany, **3** Institute of Clinical Neuroanatomy, Neuroscience Center, Goethe University, Frankfurt/Main, Germany, **4** Frankfurt Institute for Advanced Studies, Frankfurt am Main, Germany

* wkilb@uni-mainz.de

**Data Availability Statement:** The source code of all models and stimulation files used in the present paper can be found in ModelDB (http://modeldb.yale.edu/266823).

## Abstract

The impact of GABAergic transmission on neuronal excitability depends on the Cl$^-$-gradient across membranes. However, the Cl$^-$-fluxes through GABA$_A$ receptors alter the intracellular Cl$^-$ concentration ($[Cl^-]_i$) and in turn attenuate GABAergic responses, a process termed ionic plasticity. Recently it has been shown that coincident glutamatergic inputs significantly affect ionic plasticity. Yet how the $[Cl^-]_i$ changes depend on the properties of glutamatergic inputs and their spatiotemporal relation to GABAergic stimuli is unknown. To investigate this issue, we used compartmental biophysical models of Cl$^-$ dynamics simulating either a simple ball-and-stick topology or a reconstructed CA3 neuron. These computational experiments demonstrated that glutamatergic co-stimulation enhances GABA receptor-mediated Cl$^-$ influx at low and attenuates or reverses the Cl$^-$ efflux at high initial $[Cl^-]_i$. The size of glutamatergic influence on GABAergic Cl$^-$-fluxes depends on the conductance, decay kinetics, and localization of glutamatergic inputs. Surprisingly, the glutamatergic shift in GABAergic Cl$^-$-fluxes is invariant to latencies between GABAergic and glutamatergic inputs over a substantial interval. In agreement with experimental data, simulations in a reconstructed CA3 pyramidal neuron with physiological patterns of correlated activity revealed that coincident glutamatergic synaptic inputs contribute significantly to the activity-dependent $[Cl^-]_i$ changes. Whereas the influence of spatial correlation between distributed glutamatergic and GABAergic inputs was negligible, their temporal correlation played a significant role. In summary, our results demonstrate that glutamatergic co-stimulation had a substantial impact on ionic plasticity of GABAergic responses, enhancing the attenuation of GABAergic inhibition in the mature nervous systems, but suppressing GABAergic $[Cl^-]_i$ changes in the immature brain. Therefore, glutamatergic shift in GABAergic Cl$^-$-fluxes should be considered as a relevant factor of short-term plasticity.

**Funding:** This research was funded by grants of the Deutsche Forschungsgemeinschaft to WK (KI-835/3) and to HJL (CRC 1080, A01), by the BMBF grant to PJ (No. 031L0229) and by funds from the von Behring Röntgen Foundation to PJ. The funders had no role in study design, data collection and analysis, decision to publish, or preparation of the manuscript.

**Competing interests:** The authors have declared that no competing interests exist

## Author summary

Information processing in the brain requires that excitation and inhibition are balanced. The main inhibitory neurotransmitter in the brain is gamma-amino-butyric acid (GABA). GABA actions depend on the $Cl^-$-gradient, but activation of ionotropic GABA receptors causes $Cl^-$-fluxes and thus reduces GABAergic inhibition. Here, we investigated how a coincident membrane depolarization by excitatory glutamatergic synapses influences GABA-induced $Cl^-$-fluxes using a biophysical compartmental model of $Cl^-$ dynamics, simulating either simple or realistic neuron topologies. We demonstrate that glutamatergic co-stimulation directly affects GABA-induced $Cl^-$-fluxes, with the size of glutamatergic effects depending on the conductance, the decay kinetics, and localization of glutamatergic inputs. We also show that the glutamatergic shift in GABAergic $Cl^-$-fluxes is surprisingly stable over a substantial range of latencies between glutamatergic and GABAergic inputs. We conclude from these results that glutamatergic co-stimulation alters GABAergic $Cl^-$-fluxes and in turn affects the strength of GABAergic inhibition. These coincidence-dependent ionic changes should be considered as a relevant factor of short-term plasticity in the CNS.

## Introduction

Information transfer within networks of single neurons is carried out via synaptic contacts, that use different neurotransmitters acting on pre- and postsynaptic receptors. The two most important neurotransmitters in the mammalian brain are glutamate and γ-amino butyric acid (GABA), which in general exert an excitatory and inhibitory action in the postsynaptic cell, respectively [1]. Beside regulating the excitability of neuronal circuits, GABA is also required for the control of sensory integration, regulation of motor functions, generation of oscillatory activity, and neuronal plasticity [2]. The responses to GABA are mediated by metabotropic $GABA_B$ receptors and by ionotropic $GABA_A$ receptors, which are ligand-gated anion-channels with a high $Cl^-$ permeability and a lower $HCO_3^-$ permeability [1]. The GABAergic effects therefore depend mainly on the $Cl^-$ gradient across the neuronal plasma membrane [1, 3]. The typical inhibitory action of GABA requires a low intracellular $Cl^-$ concentration ($[Cl^-]_i$), which is typically maintained by the action of a $Cl^-$ extruder termed KCC2 [3]. During early development $[Cl^-]_i$ is maintained at high levels by active $Cl^-$ accumulation via the $Na^+$-$K^+$-$2Cl^-$-Symporter NKCC1, rendering GABA responses depolarizing and under certain conditions excitatory [4–6]. Such excitatory GABAergic responses contribute to the generation of spontaneous neuronal activity, which is essential for the correct maturation of nervous systems [7–9].

However, the $Cl^-$-fluxes through $GABA_A$ receptors, which underlie GABAergic currents, change $[Cl^-]_i$ and thus temporarily affect the amplitude of subsequent GABAergic responses, a process termed ionic-plasticity [3, 10, 11]. Such activity-dependent $[Cl^-]_i$ transients have been observed in various neurons [12–19]. Ionic plasticity plays an important role for physiological functions [20–22] as well as for pathophysiological processes [23, 24]. Theoretical assumptions and computational studies indicate that the amount and duration of $[Cl^-]_i$ ionic plasticity directly depends on the relation between $Cl^-$ influx and the capacity of $Cl^-$ extrusion systems [3, 24–28]. In addition, the size and geometrical structure of the postsynaptic compartments critically affect the magnitude, duration and dimensions of $[Cl^-]_i$ changes upon GABAergic activation [27, 29]. Further analyses also revealed that the membrane resistance, the kinetics of GABAergic responses and the stability of bicarbonate gradients affect the magnitude and duration of $[Cl^-]_i$ changes [12, 27, 30].

Another factor that directly influences the amount of $GABA_A$ receptor-mediated $Cl^-$ fluxes is a coincident membrane depolarization [11, 27, 31]. Accordingly, recent in-vitro and in-silico studies demonstrated that coincident glutamatergic depolarization profoundly augments the $GABA_A$ receptor mediated $Cl^-$ fluxes [32, 33]. However, to our knowledge it has not been analyzed how the interdependency between glutamatergic and GABAergic inputs affects the magnitude and the spatiotemporal properties of activity dependent $[Cl^-]_i$ changes.

In order to determine the influence of coincident glutamate stimulation on $GABA_A$ receptor-induced $[Cl^-]_i$ transients, we utilized a computational model of $[Cl^-]_i$ homeostasis in the NEURON environment [25, 30]. Using a simple ball-and-stick geometry, we were able to show that the conductance and decay kinetics of glutamatergic responses directly affect the size of GABAergic $[Cl^-]_i$ changes, with a complex spatiotemporal dependency between glutamatergic and GABAergic inputs. Furthermore, we employed the model to uncover the contribution of coincident glutamatergic activity on the activity-dependent $[Cl^-]_i$ transients during simulated giant-depolarizing potentials (GDPs) [17, 30]. GDPs represent correlated spontaneous network events crucial for the development of neuronal circuits [34–36]. These simulations demonstrate for the first time that coincident glutamatergic stimulation with realistic parameters substantially modifies the $GABA_A$ receptor-induced $[Cl^-]_i$ changes.

## Results

In order to analyze how co-activation of depolarizing neurotransmitter receptors affects the $[Cl^-]_i$ transients evoked by $GABA_A$ receptor activation, we used a previously established biophysical model of $Cl^-$ dynamics in the NEURON environment [17, 25]. The model allows for realistic simulations of activity-dependent transmembrane $Cl^-$ fluxes, intracellular $Cl^-$ diffusion and corresponding intracellular $Cl^-$ accumulation or depletion (see Methods). In a first step we computed the GABA-induced $[Cl^-]_i$ changes in a ball-and-stick model with a single dendrite (Fig 1A), in order to enable a detailed mechanistic understanding of the interaction between a single or a small group of $GABA_A$ receptor-mediated and glutamate receptor-mediated synaptic inputs (Figs 1–6). Subsequently we utilized a model of a reconstructed CA3 pyramidal neuron [17, 30] stimulated by a large number of inputs to compute a more realistic estimation of how the $GABA_A$ receptor-mediated $[Cl^-]_i$ transients in neurons are influenced by correlated co-activation of glutamate receptors (Figs 7–9).

### Effect of GABA and AMPA co-activation for a weak focal synaptic activation

First we analyzed the effect of glutamate receptor co-activation on the $GABA_A$ receptor-induced $[Cl^-]_i$ transients. It is important to emphasize that simulations of single or a low number of synaptic inputs necessarily led to small absolute effects of glutamatergic activation on GABAergic $[Cl^-]_i$ or membrane voltage ($E_m$) changes (Figs 1–6), but nevertheless allowed for systematic parameter scans important for the detailed understanding of glutamatergic effects (see below). For quantitatively stronger effects during more realistic GABAergic and glutamatergic distributed input activation see Sections 2.4 and 2.5.

Simulation of a single GABA synapse using parameters determined in-vitro for spontaneous GABAergic postsynaptic currents (PSCs) ($g_{GABA}$ = 0.789 nS, $\tau_{GABA}$ = 37 ms) [17], induced $[Cl^-]_i$ changes (Fig 1B and 1C) that depended on the initial $[Cl^-]_i$ ($[Cl^-]_i^0$), as has been shown before [17]. At low $[Cl^-]_i^0$ the inwardly directed driving force ($DF_{GABA} = E_m—E_{GABA}$) caused a $Cl^-$-influx and thus a membrane hyperpolarization. At higher $[Cl^-]_i^0$ of 25 mM the outwardly directed $DF_{GABA}$ caused a $Cl^-$-efflux and thus a membrane depolarization, while at an $[Cl^-]_i^0$ of 15 mM, which is close to the $[Cl^-]_i$ given by a passive distribution at -60 mV, only a slight $[Cl^-]_i$

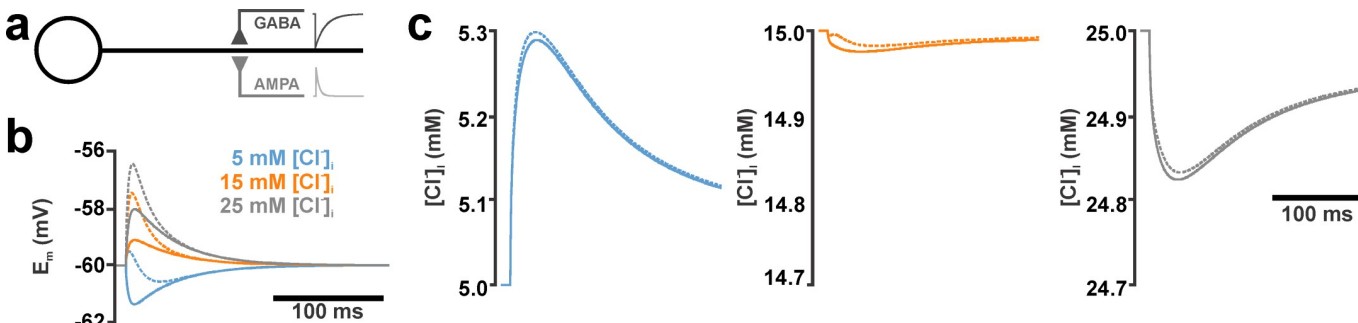

**Fig 1. Effect of AMPA co-activation on GABA_A receptor mediated [Cl⁻]_i transients for a single GABAergic and glutamatergic input in the ball-and-stick neuronal model.** (a) Schematic illustration of the compartmental model. Both GABA and AMPA synapse were located in the middle of the dendrite. The inset traces represent schematic illustrations of AMPA- and GABA-receptor mediated currents. (b) Membrane potential ($E_m$) changes induced at the site of the GABAergic synapse by a single GABAergic stimulation ($g_{GABA}$ = 0.789 nS, $\tau_{GABA}$ = 37 ms) at 5 mM (red), 15 mM (green) and 25 mM [Cl⁻]_i (blue). Solid lines represent the effect upon an isolated GABAergic stimulation, while dashed lines represent responses upon AMPA co-stimulation ($g_{AMPA}$ = 0.305 nS, $\tau_{AMPA}$ = 11 ms). (c) [Cl⁻]_i transients induced at the site of the GABAergic synapse by the isolated GABAergic stimulation (solid lines) or GABA/AMPA co-stimulation (dashed lines). Parameters as in (b). Note that under these conditions AMPA co-activation slightly enhances the [Cl⁻]_i increase at 5 mM, while it slightly attenuates the [Cl⁻]_i decline at higher [Cl⁻]_i concentrations.

efflux was induced (Fig 1B and 1C). The simultaneous co-activation with a simulated single glutamate synapse, using parameters that were determined in-vitro for spontaneous glutamatergic PSCs [17]) and resemble the properties of the AMPA subtype of glutamate receptors (*g_{AMPA} = 0.305 nS, τ_{AMPA} = 11 ms*), imposed an additional depolarizing drive to the $E_m$ responses (Fig 1B, dashed lines). Thereby it caused slight changes in the activity-dependent [Cl⁻]_i transients towards higher concentration, thus enhancing the [Cl⁻]_i increase at low [Cl⁻]_i⁰ and attenuating the [Cl⁻]_i decrease at higher [Cl⁻]_i⁰ (Fig 1C, dashed lines).

Since physiological and pathophysiological activity is typically characterized by neurotransmitter release from several release sites [37, 38], we next analyzed the effect of varying $g_{AMPA}$ on the GABA_A receptor -mediated [Cl⁻]_i transient. As expected, these simulations showed that GABA_A receptor-induced Cl⁻ fluxes depended on the [Cl⁻]_i⁰ (Fig 2A black trace). However, the Cl⁻ fluxes were systematically shifted towards Cl⁻ influx in the presence of AMPA receptor-mediated inputs (Fig 2A and 2B). At a $g_{AMPA}$ of 305 nS and 30.5 nS (corresponding to 100x and 1000x glutamatergic PSCs) a consistent Cl⁻ influx was induced even at high [Cl⁻]_i (Fig 2A and 2B). At a $g_{AMPA}$ of 0.305 nS (Fig 2C and 2E) and 3.05 nS (Fig 2D and 2E) bimodal effects, consisting of an initial influx followed by an efflux, were observed within a limited [Cl⁻]_i range. These biphasic responses were caused by the fact that the membrane depolarization only temporarily exceeds $E_{Cl}$ (Fig 2C and 2D). The maximal values of Cl⁻ in- and efflux are displayed in separate panels in Fig 2E and 2F.

A systematic analysis of the direction of Cl⁻ fluxes at various $g_{AMPA}$ revealed that the GABA_A receptor-mediated Cl⁻ fluxes show a logarithmic dependency on $g_{AMPA}$, resulting in sigmoidal curves in a monologarithmic plot (Fig 2F). Only at low [Cl⁻]_i concentrations, which permit a Cl⁻ influx even without AMPA co-stimulation, a monotonous increase in the Cl-fluxes were induced by increasing $g_{AMPA}$ (Fig 2E and 2F). When [Cl⁻]_i was above 13 mM (corresponding to $E_{Cl}$ above -60 mV) addition of $g_{AMPA}$ caused biphasic Cl- fluxes (Fig 2E and 2F).

In summary, these results indicate that co-activation of AMPA receptors enhances Cl⁻ influx at low [Cl⁻]_i and attenuates Cl⁻ efflux at high [Cl⁻]_i. with the possibility to even revert the latter to Cl⁻ influx. Already relatively low, physiologically relevant $g_{AMPA}$ values representing 10–100 spontaneous postsynaptic events, are sufficient to substantially modulate GABA_A receptor mediated Cl⁻ fluxes.

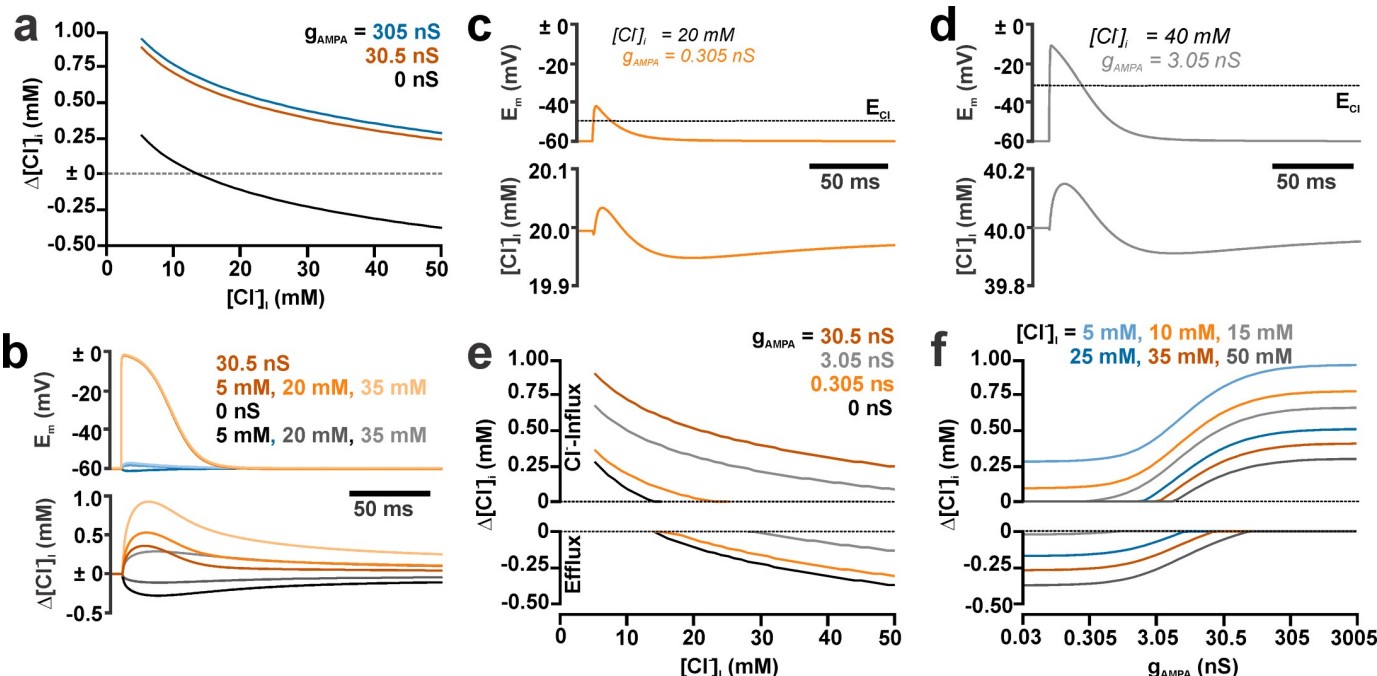

**Fig 2. Effect of $g_{AMPA}$ on the $[Cl^-]_i$ transients induced by $GABA_A$ and AMPA receptor co-activation.** (a) $GABA_A$ receptor-induced $[Cl^-]_i$ changes simulated at different $[Cl^-]_i^0$ in the absence (black trace) and the presence of AMPA receptor co-activation with high $g_{AMPA}$ as indicated in the plot. (b) Membrane depolarization (upper panel) and $[Cl^-]_i$ (lower panel) at three different $[Cl^-]_i^0$, as indicated in the plot, in the presence and absence of AMPA receptor co-activation. Note that this co-activation systematically shifted the $[Cl^-]_i$ transients towards higher $[Cl^-]_i$. (c) Membrane depolarization (upper panel) and $[Cl^-]_i$ (lower panel) at a $[Cl^-]_i^0$ of 20 mM and a $g_{AMPA}$ of 0.305 nS. Note the biphasic $[Cl^-]_i$ transient and that the $Cl^-$ influx was limited to the interval when Em is above $E_{Cl}$ (dashed line). (d) As in c but for 40 mM and $g_{AMPA}$ of 3.05 nS. (e) Dependency of the $GABA_A$ receptor-induced $[Cl^-]_i$ changes on the $[Cl^-]_i^0$ at different $g_{AMPA}$, as indicated in the plot. The upper plot represents the maximal $Cl^-$ influx and the lower plot the maximal $Cl^-$ efflux. (f) $g_{AMPA}$ dependency of the $GABA_A$ receptors induced $[Cl^-]_i$ changes at different $[Cl^-]_i^0$ as indicated in the plot. Note the shift towards Cl- influx with increasing $g_{AMPA}$. In all simulations $g_{GABA}$ = 0.789 nS, $\tau_{GABA}$ = 37 ms and $\tau_{AMPA}$ = 11 ms.

## Influence of $\tau_{AMPA}$ on $GABA_A$ receptor-induced $[Cl^-]_i$ transients

Next we analyzed the influence of the kinetics of coincident AMPA receptor activation on the $GABA_A$ receptor-induced $[Cl^-]_i$ transients. For this purpose, we modified the decay time constant of the AMPA synapse ($\tau_{AMPA}$) using values between 1 and 100 ms. These simulations were performed under consideration of the experimentally determined values for $GABA_A$ receptor-mediated inputs either at $g_{AMPA}$ of 0.305 nS (i.e. the experimentally determined value [17]) or at $g_{AMPA}$ of 3.05 nS (to estimate the effects of a small number of coincident colocalized glutamatergic inputs). As the depolarizing effect of the AMPA synapse strictly depends on $\tau_{AMPA}$ (Fig 3A), for $\tau_{AMPA}$ values < 37 ms biphasic voltage responses were induced at low $[Cl^-]_i$ (Fig 3A). In contrast, at high $[Cl^-]_i$ (where GABAergic responses are depolarizing) the peak depolarization was enhanced. For $\tau_{AMPA}$ values > 37 ms the depolarization was virtually saturated (Fig 3B). In accordance with this observation the AMPA-mediated shift in the GABAergic $[Cl^-]_i$ transients was sensitive to $\tau_{AMPA}$ at values < 37 ms, while at values > 37ms only minimal additional effects were observed (Fig 3C and 3D). These findings indicate that the kinetics of glutamatergic synaptic inputs have a consistent effect on activity-dependent $[Cl^-]_i$ changes and that shorter glutamatergic postsynaptic potentials (PSPs) will attenuate the effect of AMPA on activity-dependent $[Cl^-]_i$ transients.

Since these results implicate a substantial influence of the time course of depolarizing events on the GABA-induced $[Cl^-]_i$ transients, we also investigated the effect of NMDA-receptors, which possess slow onset and decay kinetics, on the GABA receptor-induced $[Cl^-]_i$ transients.

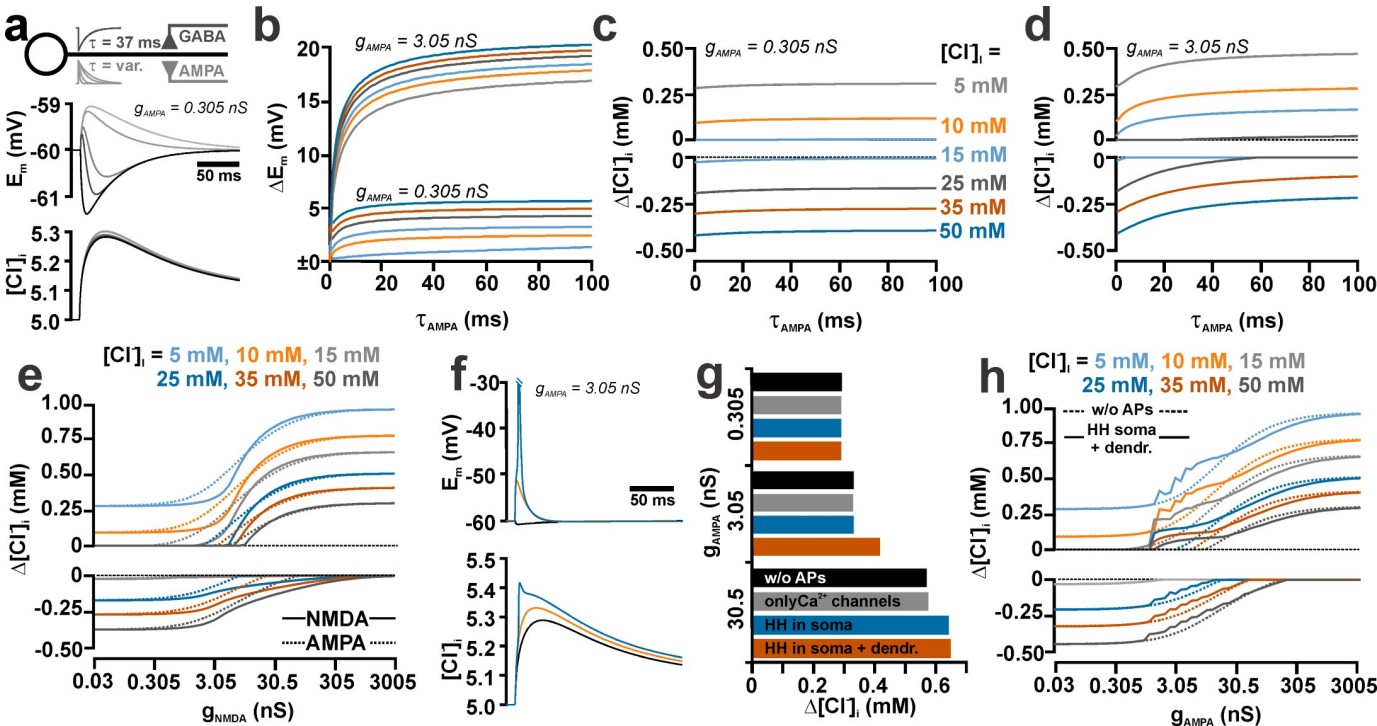

**Fig 3. Effect of $\tau_{AMPA}$ on the $[Cl^-]_i$ transients induced by $GABA_A$ and AMPA receptor co-stimulation.** (a) Schematic illustration of the compartmental model. The inset traces represent schematic illustrations of AMPA- and GABA-receptor mediated currents. The GABA synapse ($g_{GABA}$ = 0.789 nS) was stimulated with a constant $\tau_{GABA}$ of 37 ms, while $\tau_{AMPA}$ was systematically varied. The traces below the scheme displays the voltage responses (left panel) and $[Cl^-]_i$ changes induced without (black traces) or with AMPA co-stimulation ($g_{AMPA}$ = 0.305 nS) at $\tau_{AMPA}$ values of 5, 11, 37, and 50 ms (in ascending gray shades). Note the biphasic voltage responses at short $\tau_{AMPA}$. (b) Membrane potential changes induced by a AMPA/GABA co-stimulation at different $\tau_{AMPA}$ simulated for different $[Cl^-]_i$ (for color code see panel c). The upper traces were simulated using a $g_{AMPA}$ of 0.305 nS, while the lower traces represent simulation with $g_{AMPA}$ of 3.05 nS. Note that the depolarization increased with larger $\tau_{AMPA}$. (c) $[Cl^-]_i$ transients induced by AMPA/GABA co-stimulation at different $\tau_{AMPA}$ simulated with different $[Cl^-]_i$ (color coded) for a $g_{AMPA}$ of 0.305 nS. (d) As in c) but for a $g_{AMPA}$ of 3.05 nA. Note that the $[Cl^-]_i$ changes were shifted towards diminished $Cl^-$ efflux or enhanced $Cl^-$ influx with larger $\tau_{AMPA}$. (e) Effect of NMDA receptor-mediated synaptic inputs with various $g_{NMDA}$ on the $[Cl^-]_i$ transients at different $[Cl^-]_i^0$ (color coded). The dashed traces represent the responses using AMPA receptors. Note the right shift and steeper rising phase of the NMDA responses. (f) Voltage responses and $[Cl^-]_i$ changes with only GABAergic synaptic inputs (black trace) and with simultaneous AMPA inputs in a passive dendrite (orange trace) or in a dendrite with an active HH mechanism (blue trace). (g) Effect of different active properties of the dendrite on $[Cl^-]_i$ transients at a $[Cl^-]_i^0$ of 5 mM. Note the lack of influence of $Ca^{2+}$ channels and that at $g_{AMPA} \geq 3.05$ active dendritic properties augment the $[Cl^-]_i$ transients. (h) Effect of active dendritic properties on the $[Cl^-]_i$ transients at different $[Cl^-]_i^0$ (color coded). The dashed traces represent the responses in the passive model.

These simulations with an established model for the NMDA receptor [39] revealed that co-activation of NMDA receptors ($\tau_{NMDA}$ = 500 ms) induced at high $g_{NMDA}$ $[Cl^-]_i$ transients that are similar as the ones induced by AMPA receptor co-activation (Fig 3E). However, the $[Cl^-]_i$ transients showed a steeper dependency on $g_{NMDA}$, probably reflecting the $Mg^{2+}$ block [40]. Thus at moderate $g_{NMDA}$ of ~3 nS the $[Cl^-]_i$ changes were lower for a given NMDA conductance than for a similar AMPA conductance (Fig 3E).

In addition to the long-lasting NMDA receptor mediated responses, short action potentials may also provide a depolarizing drive that enhances the GABA induced $[Cl^-]_i$ transients. To investigate their influence, we implemented an established Hodgkin-Huxley (HH) mechanism [41], either only in the soma or in the soma and the dendrite, to emulate either passively or actively back-propagating actions potentials, respectively. These simulations revealed that action potentials enhanced the effect of AMPA on the GABAergic $[Cl^-]_i$ transients (Fig 3F–3H). While at a low $g_{AMPA}$ of 0.305 nS no AP was induced and thus no additional $[Cl^-]_i$ influx was induced, at a moderate $g_{AMPA}$ of 3.05 nS only actively back-propagating APs augment the $[Cl^-]_i$ changes (Fig 3G and 3H). At higher $g_{AMPA}$ also somatic APs, that spread passively

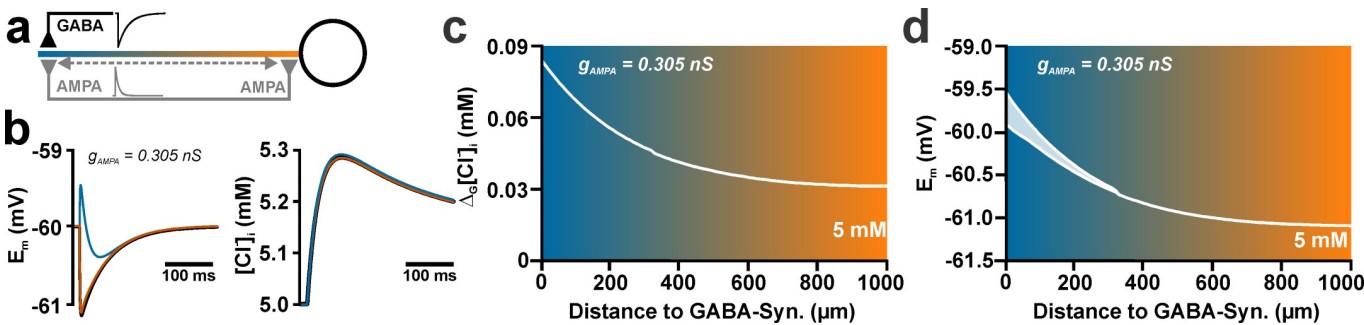

**Fig 4. Effect of the spatial relation between individual GABA and AMPA synapses on the activity-dependent $[Cl^-]_i$ transients.** (a) Schematic illustration of experimental conditions. The inset traces represent schematic illustrations of AMPA- and GABA-receptor mediated currents. (b) Voltage responses (left panel) and $[Cl^-]_i$ changes (right panel) induced without (black traces) or with AMPA co-stimulation ($g_{AMPA} = 0.305$ nS) with the AMPA synapse located at the site of the GABA synapse (blue) or close to the soma (orange). $[Cl^-]_i^0$ was 5 mM in these experiments. Note that the depolarization and $[Cl^-]_i$ responses are slightly diminished if the AMPA synapse was distant to the GABA synapse. (c) Effect of the distance between AMPA and GABA synapses on the additional $[Cl^-]_i$ influx induced by AMPA co-stimulation, as compared to a GABA stimulation without AMPA ($\Delta_G[Cl^-]_i$). Note the exponential decay of $\Delta_G[Cl^-]_i$ with increasing distance between GABA and AMPA synapses. (d) Effect of the distance between AMPA and GABA synapses on the resulting membrane depolarization. The upper and lower lines represent the maximal de- and hyperpolarizing effects for biphasic responses; above 400 μm only a monophasic hyperpolarization occurred.

(electrotonically) into the dendritic compartment, were able to augment the $[Cl^-]_i$ transients (Fig 3G). The addition of voltage gated $Ca^{2+}$ channels [41] in the dendrite had no effect on the $[Cl^-]_i$ transients (Fig 3G).

## Spatial and temporal constrains of AMPA receptor-mediated shift in GABAergic $[Cl^-]_i$ transients for a weak focal synaptic activation

To analyze how the spatial distance between the AMPA and the GABA synapse influences the $GABA_A$ receptor-induced $[Cl^-]_i$ transients, we systematically moved the AMPA synapse along the dendrite from the somatic (0%) to the distal end (100%) (Fig 4A and 4B) using the experimentally determined parameters for AMPA and GABA inputs ($g_{GABA} = 0.789$ nS, $\tau_{GABA} = 37$ ms, $g_{AMPA} = 0.305$ nS, $\tau_{AMPA} = 11$ ms [17]). These experiments revealed that the membrane potential change depends on the location of the AMPA synapse. The maximal amplitude of the positive shift in $E_m$ induced by the AMPA co-stimulation decreased substantially at distant positions (Fig 4B), due to the leak conductance shunting the synaptic currents (i.e. dendritic filtering). In accordance with these smaller depolarizing $E_m$ shifts, the $[Cl^-]_i$ changes were slightly decreased with increasing distance between AMPA and GABA synapses (Fig 4B). To quantify this effect, we calculated the amount of additional $[Cl^-]_i$ changes induced by AMPA co-stimulation ($\Delta_G[Cl^-]_i$) by subtracting the $[Cl^-]_i$ changes at GABA stimulation from the $[Cl^-]_i$ changes at AMPA/GABA co-stimulation. These simulations revealed that the decrease of $\Delta_G[Cl^-]_i$ with increasing distance between both synapses could be perfectly fitted ($R^2 = 0.023$) with a monoexponential function, using a decay length constant ($\lambda$) of 248 μm (Fig 4C). This is exactly the $\lambda$ value determined for the attenuation of the peak voltage responses (Fig 4D) and reflects the exponential decay of voltage in a linear cable with homogeneous membrane resistance. These observations indicate that the distance between GABAergic and glutamatergic synapses is a relevant factor determining the effects of co-activation on the activity-dependent $[Cl^-]_i$ transients.

To analyze how the temporal relation between the AMPA- and the GABA-mediated activity influences the $GABA_A$ receptor-induced $[Cl^-]_i$ transients we next systematically varied the delay between the AMPA and the GABAergic stimulus from -49 ms (i.e. AMPA before GABA) to +100 ms (i.e. AMPA after GABA) and determined the impact of this latency shift

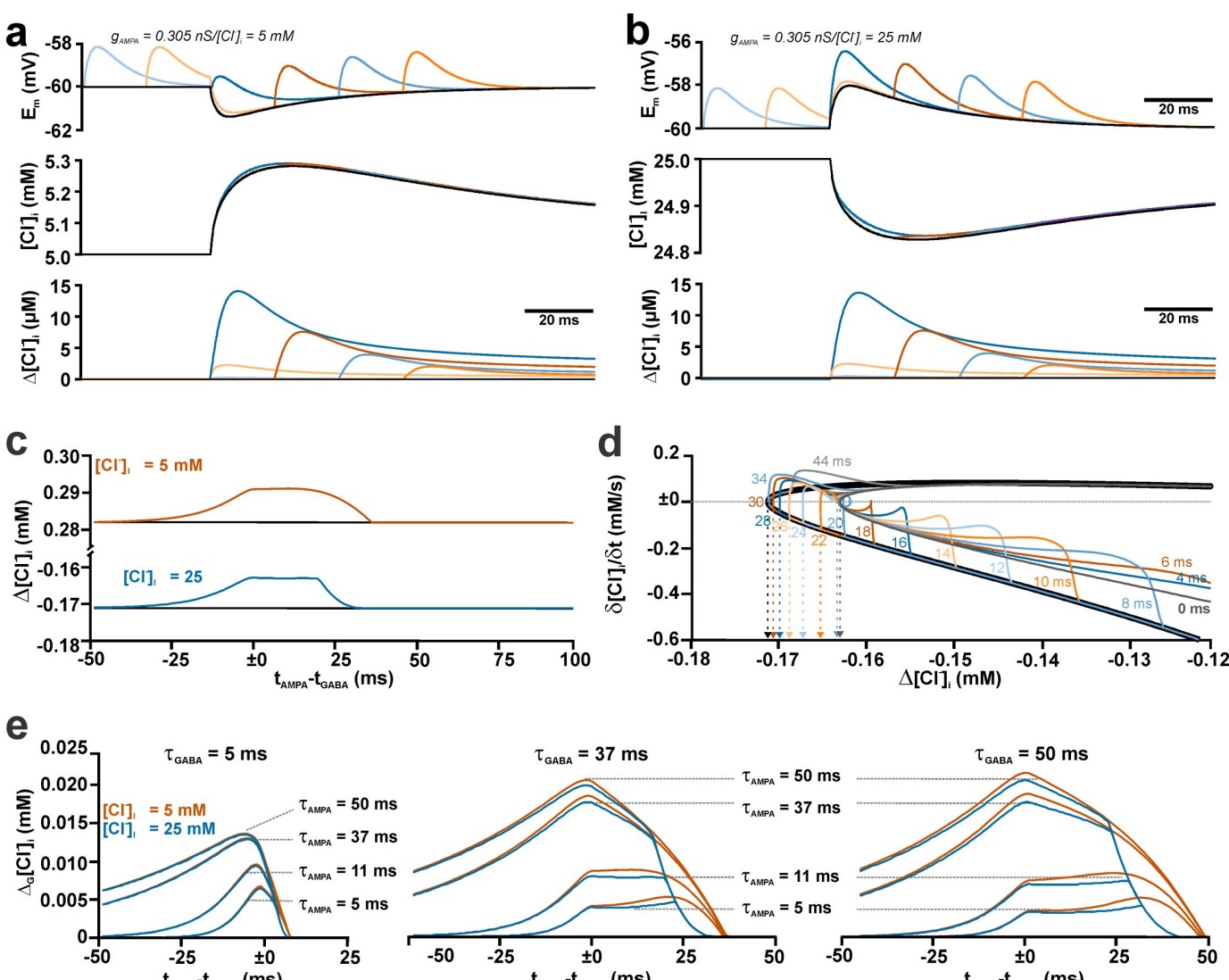

**Fig 5. Effect of the temporal relation between single dendritic GABA and AMPA synapses on the activity-dependent [Cl⁻]ᵢ transients.** (a) Voltage (upper panel) and [Cl⁻]ᵢ traces (lower panel) of [Cl⁻]ᵢ changes induced by GABAergic stimulation without AMPA (black trace) or with AMPA stimulation at different latencies (colored traces) at a [Cl⁻]ᵢ⁰ of 5 mM. (b) Same as in (a) for a [Cl⁻]ᵢ⁰ of 25 mM. (c) Plot of the peak [Cl⁻]ᵢ change at various latencies between GABA and AMPA inputs for a [Cl⁻]ᵢ⁰ of 5 mM (orange trace) and 25 mM (blue trace). The black lines represent the [Cl⁻]ᵢ change induced by stimulation of GABA synapses only. Note the obvious "plateau"-like phases in the [Cl⁻]ᵢ changes that were additionally induced by AMPA co-stimulation (Δ_G[Cl⁻]ᵢ). (d) Phase plane plot of the activity-dependent [Cl⁻]ᵢ transients (efflux) at a [Cl⁻]ᵢ⁰ of 25 mM. The black line represents the trajectory of a pure GABA stimulation. Note that all trajectories of 0 to 20 ms latency curves converged and crossed the 0 y-axis value (∂[Cl⁻]ᵢ /∂t = 0 mM/s; dashed line) at a less negative x-axis value (Δ[Cl⁻]ᵢ) than the crossings of the >22 ms latency curves. This indicates that 0–20 ms latencies similarly and robustly decrease the peak GABA-induced efflux (i.e. 0 y-axis crossing of the pure GABA black line). See main text for further description. (e) Dependency of Δ_G[Cl⁻]ᵢ on the latency between AMPA and GABA stimulation simulated for different τ_GABA and τ_AMPA. Note the plateau-like phases occurring for τ_GABA ≥ 37 ms at short τ_AMPA.

on the [Cl⁻]ᵢ transients (Fig 5A and 5B). The parameters for AMPA and GABA inputs were identical to the parameters used before (g_GABA = 0.789 nS, τ_GABA = 37 ms, g_AMPA = 0.305 nS, τ_AMPA = 11 ms) and both synapses were located at the same position. These simulations revealed that the additional effect of AMPA co-stimulation on the [Cl⁻]ᵢ transients (Δ_G[Cl⁻]ᵢ) became maximal when GABA and AMPA stimulus were provided simultaneously (Fig 5C). Surprisingly, this AMPA effect on Δ_G[Cl⁻]ᵢ remained stable for a latency of ≈20 ms, before it rapidly declined (Fig 5C). To provide a mechanistic explanation for this plateau phase, we next

plotted the rate of $[Cl^-]_i$ changes versus the absolute value of the $[Cl^-]_i$ change (Fig 5D). This phase plane plot illustrates that the trajectories of all AMPA stimulations with a latency between 0 and 20 ms converged with the trajectory of the 0 ms latency stimulus (purple line), thus reaching identical minimal $[Cl^-]_i$ values (obtained at the intersection with the y-axis value $\partial[Cl^-]_i/\partial t = 0$ mM/s). Latencies between 22 and 34 ms provided a gradual decline in the minimal $[Cl^-]_i$. For latencies > 34 ms the additional AMPA-mediated reduction of the $Cl^-$-efflux was initiated after a minimal $[Cl^-]_i$ was reached, therefore these stimulations did not provide a $[Cl^-]_i$ decrease exceeding the $[Cl^-]_i$ decrease mediated by a pure GABA stimulation (black line).

To investigate how the duration of GABA and AMPA receptor-mediated currents influence this complex time course of $[Cl^-]_i$ transients, we next systematically varied $\tau_{GABA}$ (5 ms, 37, ms 50 ms) as well as $\tau_{AMPA}$ (5 ms, 11 ms, 37, ms 50 ms) and determined $\Delta_G[Cl^-]_i$ (Fig 5E). These simulations revealed that, in accordance with the previous results, $\Delta_G[Cl^-]_i$ increased at longer $\tau_{AMPA}$ for all three $\tau_{GABA}$ tested. While at a short $\tau_{GABA}$ of 5 ms no plateau phase was observed, such a plateau occurred for $\tau_{GABA}$ of 37 ms and 50 ms with short $\tau_{AMPA}$ values of 5 ms or 11 ms (Fig 5E). However, in contrast to our hypothesis that such a plateau phase occurred when $\tau_{AMPA}$ was shorter than $\tau_{GABA}$, for a $\tau_{GABA} \geq 37$ ms and $\tau_{AMPA} \geq 37$ ms $\Delta_G[Cl^-]_i$ steadily declined after the maximal value was obtained at simultaneous AMPA/GABA stimulation. Another finding of these simulations that appears counter-intuitive is the fact that for short $\tau_{AMPA}$ of 5 ms and 11 ms the peak $\Delta_G[Cl^-]_i$ decreases when $\tau_{GABA}$ increased from 5 ms to 37 ms (Fig 5E).

To investigate these issues in detail, we next systematically varied $\tau_{GABA}$ between 5 and 50 ms using fixed $\tau_{AMPA}$ values of 5, 11, and 37 ms and determined $\Delta_G[Cl^-]_i$ at different AMPA/GABA latencies. In these simulations we were able to identify a complex dependency between $\Delta_G[Cl^-]_i$ and the relation between $\tau_{GABA}$ and $\tau_{AMPA}$ (Fig 6A–6C). For $\tau_{AMPA}$ of 5 and 11 ms the influence of the AMPA/GABA latency on $\Delta_G[Cl^-]_i$ changed from a "peak"-like pattern to a "plateau"-like phase, in which the maximal $\Delta_G[Cl^-]_i$ remained rather constant for a progressively longer latency interval between AMPA and GABA stimulation. This "plateau"-like phase occurred under conditions when $\tau_{GABA}$ is at more than about 3 times larger than $\tau_{AMPA}$ (Fig 6A and 6B). For $\tau_{AMPA}$ of 37 ms this "plateau"-like phase was not reached, but a phase with linearly decreasing $[Cl^-]_i$ shifts became visible (Fig 6C). In any way, the maximal AMPA-dependent $[Cl^-]_i$ shift was observed at $\tau_{GABA}$ of 11 ms, 18 ms and 47 ms for $\tau_{AMPA}$ of 5 ms, 11 ms and 37 ms, respectively (red plots in Fig 6A), reproducing the observation that prolonging $\tau_{GABA}$ can reduce $\Delta_G[Cl^-]_i$. From these results it can be concluded, first, that the effect of AMPA co-stimulation on $\Delta_G[Cl^-]_i$ depends critically on the timing between both inputs, second, that $\Delta_G[Cl^-]_i$ is maximal when $\tau_{GABA}$ is slightly larger than $\tau_{AMPA}$, and, third, that a "plateau"-like interval of stable $\Delta_G[Cl^-]_i$ occurred when $\tau_{GABA}$ is at least 3 times larger than $\tau_{AMPA}$.

Next, we investigated how the time course of $E_m$ changes contributes to this complex dependency. For this purpose, we used identical stimulation parameters ($g_{GABA} = 0.789$ nS, $\tau_{GABA} = 37$ ms, $g_{AMPA} = 0.305$ nS, $\tau_{AMPA} = 11$ ms, $[Cl^-]_i^0 = 25$ mM) and modified the time course of the voltage traces by changing the membrane time constant (S1 Fig). These simulations revealed that reducing the membrane time constant, which enhanced the onset kinetics of AMPA and GABA receptors, as well as the decay of AMPA receptor-mediated voltage responses, had only a minor effect on the observed $[Cl^-]_i$ changes (S1A Fig). The trajectories at different latencies converge virtually in the same manner as at physiological conditions (S1B Fig). On the other hand, a substantial prolongation of the membrane time constant slowed the kinetics of both AMPA and GABA-receptor dependent depolarizations (S1C Fig) and led to significantly different behavior of $[Cl^-]_i$ changes: The delayed GABAergic depolarization increased $\Delta[Cl^-]_i$ when GABAergic input was stimulated alone (black trace). In addition, the

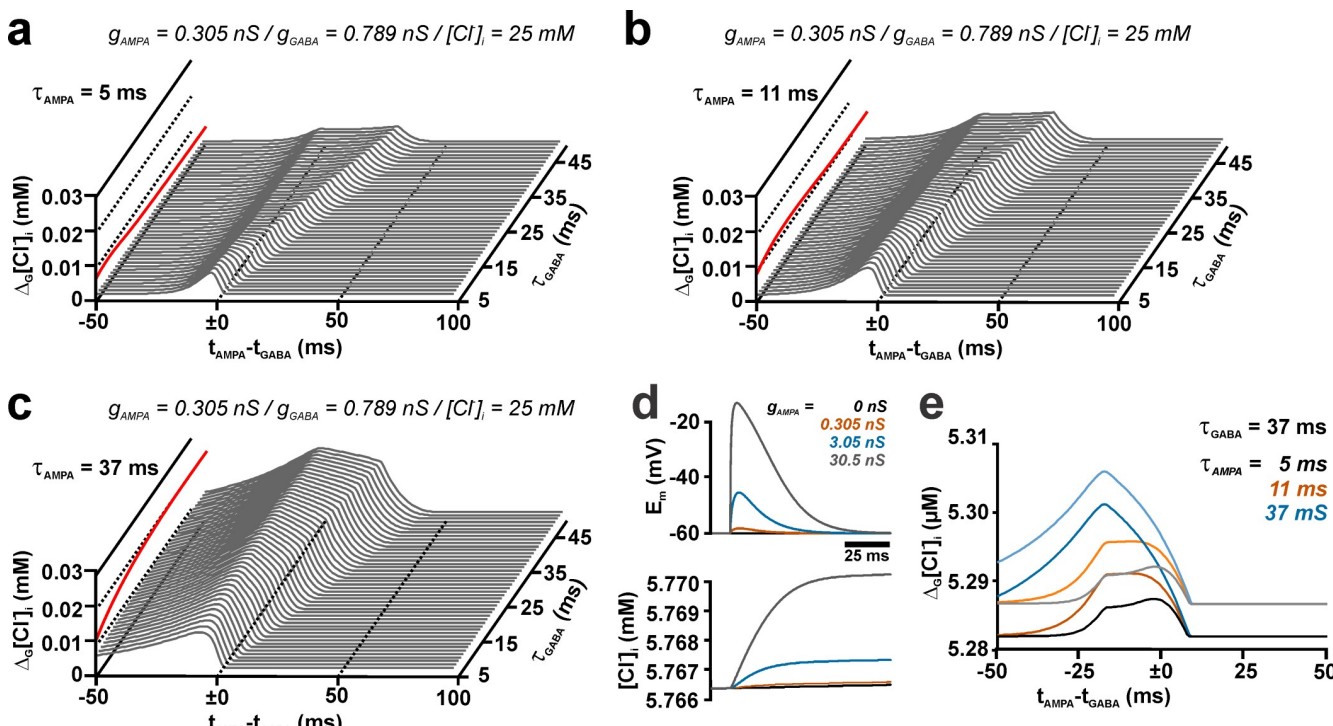

**Fig 6. Effect of the relation between kinetics of individual dendritic GABA and AMPA synapses on the activity-dependent $[Cl^-]_i$ transients at different temporal relations between both synapses.** (a-c) Amplitude of the additional $[Cl^-]_i$ influx induced by AMPA co-stimulation (compared to a GABA stimulation without AMPA) at different intervals between AMPA and GABA activation. Simulations were performed for 3 different $\tau_{AMPA}$ of 5 ms (a), 11 ms (b), and 37 ms (c) with $\tau_{GABA}$ varying between 5 ms and 50 ms. Note the evolution of a "plateau-like" phase with increasing $\tau_{GABA}$. The peak $[Cl^-]_i$ change was displayed as red trace on the left face of the plots. Note that for $\tau_{AMPA}$ of 5 ms and 11 ms, increasing $\tau_{GABA}$ above 10 ms and 17 ms reduced the $[Cl^-]_i$ shift induced by AMPA co-stimulation. (d) Voltage (upper panel) and $[Cl^-]_i$ traces induced by AMPA conductance activation ($\tau_{AMPA}$ = 11 ms) in the presence of a tonic GABAergic conductance of 8.75 $nS/cm^2$. Please note the minimal $[Cl^-]_i$ changes induced under this condition. (e) Activity-dependent $[Cl^-]_i$ transients at different intervals between AMPA and GABA activation in the absence (dark colors) and presence of the tonic GABAergic conductance (light colors). Note that the amount of activity dependent $[Cl^-]_i$ changes was unaffected by the tonic current, despite a shift in the basal $[Cl^-]_i$ under these conditions.

maximum amount of $\Delta_G[Cl^-]_i$ (i.e. the difference between the 0 ms latency trajectory and the GABA only trajectory) was smaller. More importantly, the trajectories for latencies > 4 ms did no longer converge with the trajectory of simultaneous AMPA/GABA stimulation (purple line), which resulted in a steady decline of $\Delta_G[Cl^-]_i$ for all conditions in which AMPA was stimulated after the GABA input. In summary, these results indicate a complex interplay between the membrane time constant and the kinetics of AMPA- and GABA-mediated synaptic events. Only in cases in which the kinetics of the receptors dominate the time course of membrane responses, the complex, plateau-like dependency of $\Delta_G[Cl^-]_i$ on the latency between GABA and AMPA inputs was observed.

While in these models the background conductance was set by a purely passive conductance, it had been demonstrated that tonic $Cl^-$ currents considerably contribute to the background conductance [42]. Therefore, we additionally implemented a tonic background conductance of 8.75 $nS/cm^2$ [43] and reduced the passive conductance to 0.99125 $mS/cm^2$, to maintain the input resistance of 188.2 M$\Omega$. These simulations revealed that under this condition a constant increase in the basal $[Cl^-]_i$ occurred. In the presence of this tonic $[Cl^-]_i$ conductance activation of AMPA-mediated synapses led to $[Cl^-]_i$ changes that depended on $g_{AMPA}$ and $\tau_{AMPA}$ (S2 Fig). However, at physiologically relevant values for AMPA receptor-mediated inputs ($g_{AMPA}$ = 0.305 nS and $\tau_{AMPA}$ = 11 ms), only a minimal additional $[Cl^-]_i$ increase by

0.1 μM was induced, which slightly increased to 0.85 μM and 3.74 μM at gAMPA of 3.05 nS and 30.05 nS, respectively (Figs 6D and S2). Accordingly, addition of a tonic GABAergic current had no effect on the $[Cl^-]_i$ changes induced by phasic GABAergic inputs, despite a slight shift in the basal $[Cl^-]_i$ (Fig 6E).

### Influence of gAMPA and τAMPA on the GABAergic $[Cl^-]_i$ transients in a morphologically realistic neuronal model with multiple distributed synaptic inputs

To investigate in a more physiological setting how the interference between $GABA_A$ and AMPA receptors influence the resulting $[Cl^-]_i$ transients, we implemented GABA and AMPA synapses in a morphologically realistic model of an immature CA3 pyramidal neuron (Fig 7A and 7B), using morphology and membrane parameters determined in in-vitro experiments [17]. In the majority of simulations, we applied experimentally estimated correlated GABAergic and glutamatergic activity recorded during giant depolarizing potentials (GDPs, [17], see Fig 7C). GDPs are highly relevant network events in the immature hippocampus that have been shown to cause substantial $[Cl^-]_i$ changes [8, 17, 35, 44]. Therefore, the use of these neurons and their activity patterns allows to simulate ionic plasticity in a morphologically and physiologically relevant setting. This computational model, which incorporates basic morphological and biophysical properties of hippocampal neurons, generated complex trajectories of $[Cl^-]_i$ within individual dendrites during a simulated GDP (Fig 7D). These $[Cl^-]_i$ changes in the individual neurites are determined by $Cl^-$-fluxes via $GABA_A$ receptors, lateral $Cl^-$ diffusion within the dendritic compartment and transmembrane $Cl^-$-transport [25, 29, 33]. For a better

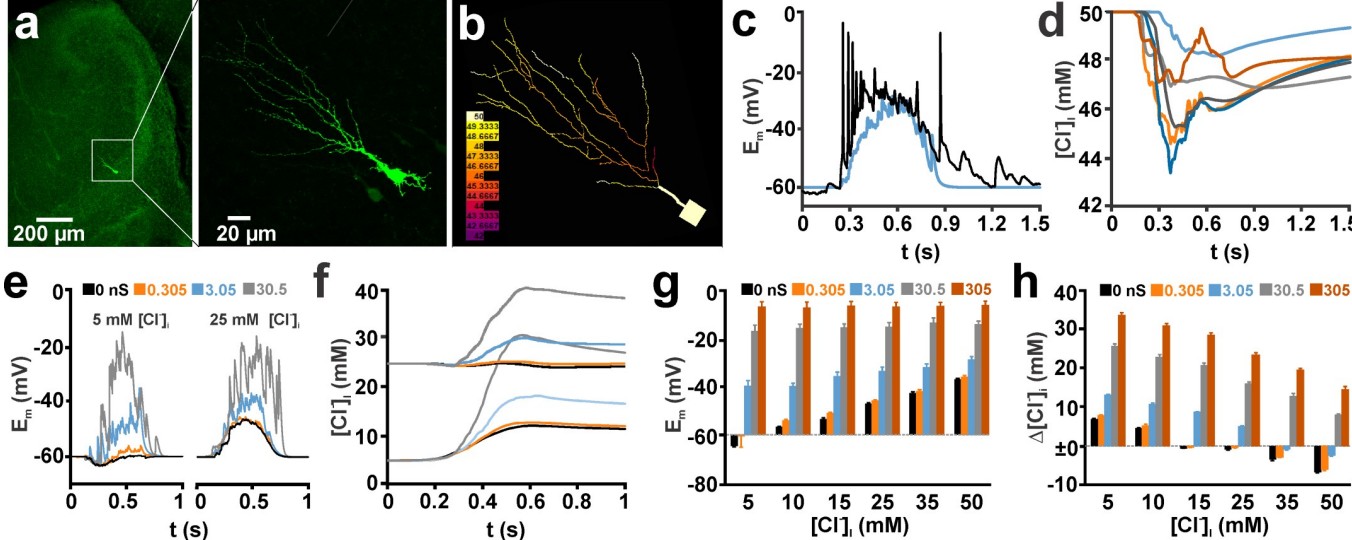

**Fig 7. Effect of AMPA receptor co-stimulation on $GABA_A$ receptor-induced $[Cl^-]_i$ transients in a morphologically realistic neuronal model stimulated with multiple spatially distributed synaptic inputs replicating GDP-like depolarizations (GDPs are experimentally observed giant depolarizing potentials in immature neurons).** (a) Microfluorimetric image of a biocytin-filled CA3 pyramidal neuron stained during an electrophysiological recording. (b) Morphological representation of this neuron in the NEURON environment. The colors represent the actual $[Cl^-]_i$ during a simulated GDP. (c) Recorded (black-trace) and simulated (blue trace) voltage deflections during a GDP with GABAergic synaptic inputs only. Note that no Hodgkin-Huxley spiking mechanisms were implemented in the model. (d) Time course of $[Cl^-]_i$ in the center node of 6 typical dendrites during this simulated GDP. (e) Typical voltage deflections during a GDP at a $[Cl^-]_i{}^0$ of 5 mM (left panel) and 25 mM (right panel) using different amounts of AMPA co-stimulation inputs as indicated by the color code. Note the substantial shift towards depolarized potentials at high gAMPA. (f) Time course of average dendritic $[Cl^-]_i$ at different $[Cl^-]_i{}^0$ and gAMPA (as indicated in e). Note that at a $[Cl^-]_i{}^0$ of 5 mM the maximal $[Cl^-]_i$ change was augmented by addition of 107 AMPA synapses with a gAMPA of 0.305 nS, while the $[Cl^-]_i$ decline at a $[Cl^-]_i{}^0$ of 25 mM was attenuated by this AMPA co-stimulation. (g) Statistical analysis of the voltage changes induced by simulated GDPs with the given $[Cl^-]_i{}^0$ and gAMPA. (h) Statistical analysis of $[Cl^-]_i$ changes induced by simulated GDPs with the given $[Cl^-]_i{}^0$ and gAMPA. Bars represent mean ± SD of 9 repetitions. Panel (a) used with permission from Lombardi et al. [17].

quantification and display, the average $[Cl^-]_i$ over all nodes of all dendrites was calculated at each simulated interval (compare e.g. Fig 7F), while $E_m$ was determined in the soma.

In a first set of simulations, we investigated the impact of $g_{AMPA}$ on GABAergic $[Cl^-]_i$ changes during a simulated GDP. For the simulation of a GDP, 534 GABAergic synapses were randomly distributed within the dendrites and activated stochastically to emulate the distribution of GABAergic PSCs during a GDP observed in-vitro [17] (see Materials and Methods for detail). In accordance with previous observations [17, 30], the massive GABAergic activity during a GDP led to substantial $[Cl^-]_i$ changes, with a $[Cl^-]_i$ increase by 3.94 ± 0.05 mM (n = 9 repetitions with random distribution/stimulation) at a low $[Cl^-]_i^0$ of 5 mM and a $[Cl^-]_i$ decrease by -3.78 ± 0.08 mM (n = 9) at a $[Cl^-]_i^0$ of 25 mM (black traces in Fig 7E and 7F). Adding 107 AMPA synapses with a $g_{AMPA}$ of 0.305 nS (i.e. the experimentally determined number and conductance of AMPA inputs) led to a substantial increase in the activity-dependent $[Cl^-]_i$ transients to 4.95 ± 0.07 mM for a $[Cl^-]_i^0$ of 5 mM and significantly attenuated the activity-dependent $[Cl^-]_i$ decrease at 25 mM to -2.88 ± 0.06 mM (Fig 7F red trace). A similar trend was also observed for other $[Cl^-]_i^0$ concentrations. Co-stimulation with 107 AMPA inputs increased the $[Cl^-]_i$ transients at low $[Cl^-]_i^0$ and attenuated the transients at high $[Cl^-]_i^0$ (Fig 7H). Using a larger value for $g_{AMPA}$ enhanced this impact on activity-dependent $[Cl^-]_i$ shift (Fig 7F). At $g_{AMPA}$ values of 30.5 and 305 nS at all investigated $[Cl^-]_i^0$, an increase in $[Cl^-]_i$ was induced (Fig 7H), in accordance with the depolarized $E_m$ induced by these conditions (Fig 7G). In summary, these results indicate that physiological levels of AMPA co-stimulation can significantly affect the GABA$_A$ receptor induced $[Cl^-]_i$ transients. In line with the results from the ball-and-stick model, implementation of the tonic GABAergic conductance of 8.75 nS/cm$^2$ [43] had only a negligible effect (S3 Fig). With this tonic current, the $[Cl^-]_i$ increase at physiological $g_{AMPA}$ values of 0.305 nS was augmented by only 0.046 mM at a $[Cl^-]_i^0$ of 5 mM and the $[Cl^-]_i$ decrease at a $[Cl^-]_i^0$ of 25 mM was reduced by only 0.036 mM. Even at higher $g_{AMPA}$ values the effect of tonic GABAergic conductances remained below 0.1 mM.

In the next series of simulations, we systematically altered the decay time constant $\tau_{AMPA}$ for all 107 AMPA receptor-mediated synaptic inputs during a GDP from the experimentally determined value of 11 ms to 5, 37 and 50 ms, while using $g_{AMPA}$ of 0.305 nS and the experimentally determined values for the GABAergic synaptic inputs. Although this slight shift in $\tau_{AMPA}$ had only a minor effect on the size of the voltage deflections (Fig 8A and 8C), it significantly modified the effect of AMPA co-stimulation on GABA$_A$ receptor mediated $[Cl^-]_i$ (Fig 8B and 8D). At a $[Cl^-]_i^0$ of 5 mM the enhancement of GDP-induced $[Cl^-]_i$ transient by AMPA ($\Delta_G[Cl^-]_i$) was reduced from 1.02 ± 0.06 mM (n = 9 repetitions) at $\tau_{AMPA}$ = 11 ms to 0.48 ± 0.06 mM at $\tau_{AMPA}$ = 5 ms (Fig 8B). Similarly, at a $[Cl^-]_i^0$ of 25 mM, $\Delta_G[Cl^-]_i$ was diminished by shortening $\tau_{AMPA}$ from 11 ms (0.93 ± 0.1 mM) to 5 ms (0.43 ± 0.07 mM) (Fig 8B). In contrast, prolonging $\tau_{AMPA}$ to 37 or even 50 ms had a substantial impact on the time course and amount of the voltage deflections (Fig 8A). Accordingly, the activity-dependent $[Cl^-]_i$ transients were systematically shifted toward more influx or less efflux, respectively (Fig 8B–8D). In summary, these results indicate that even slight changes in $\tau_{AMPA}$ can substantially influence the impact of AMPA co-activation on GABA$_A$ receptor-mediated $[Cl^-]_i$ transients.

In addition, we also evaluated the effect of NMDA receptors, which possess slow onset and decay kinetics, on the GABA receptor-induced $[Cl^-]_i$ transients. These simulations, using the same temporal and spatial pattern of glutamatergic inputs, but an established model for the NMDA receptor [39], revealed that co-activation of NMDA receptors ($\tau_{NMDA}$ = 500 ms) induced in general larger $[Cl^-]_i$ transients as compared to AMPA receptor co-activation (S4 Fig). While the effects were small at a $g_{NMDA}$ of 0.305 nS (1.2 ± 0.1 mM vs. 0.9 ± 0.1 mM at $[Cl^-]_i^0$ of 5 mM), substantial differences were observed at intermediate $g_{NMDA}$ values (17.5 ± 0.7 mM vs. 6.1 ± 0.3 mM at 3.05 nS or 29.3 ± 0.8 mM vs. 18.6 ± 0.7 mM at 30.5 nS) (S4 Fig).

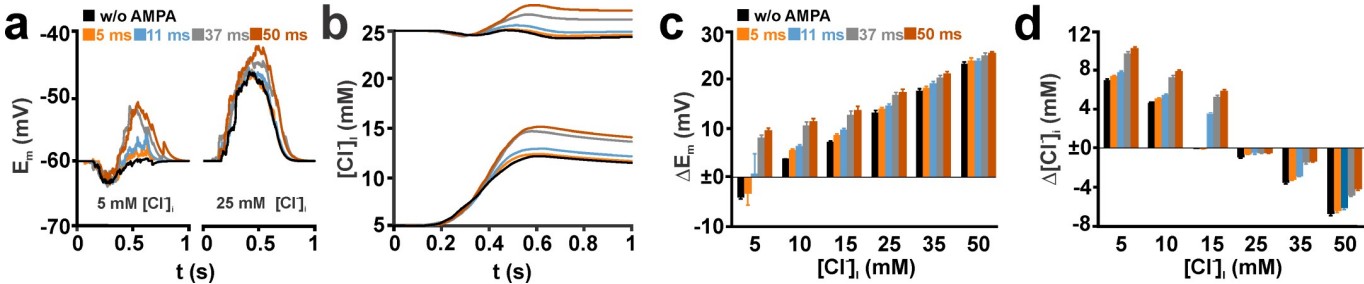

**Fig 8. Effect of the decay time of AMPA receptor-mediated currents ($\tau_{AMPA}$) on activity-dependent $[Cl^-]_i$ transients induced by multiple spatially distributed synaptic inputs replicating GDP-like depolarizations.** (a) Typical voltage traces in a simulated CA3 pyramidal neuron upon GABA stimulation (black trace) and GABA-AMPA co-stimulation with different $\tau_{AMPA}$ (color coded). Left traces represent stimulations at a $[Cl^-]_i^0$ of 5 mM, right traces at a $[Cl^-]_i^0$ of 25 mM. Note that a prolongation of $\tau_{AMPA}$ above 11 ms led to an obvious shift of the voltage response to positive values. (b) Average $[Cl^-]_i$ observed in these experiments. Note that $[Cl^-]_i$ changes upon GABA-AMPA co-stimulation were only slightly shifted at $\tau_{AMPA}$ of 5 and 11 ms, while at 37 ms and 50 ms an obvious shift towards $Cl^-$ efflux was induced. (c) Analysis of the voltage responses upon AMPA-GABA co-stimulation. (d) Analysis of the average $[Cl^-]_i$ changes upon GABA-AMPA co-stimulation. Note that for small $[Cl^-]_i^0$ longer $\tau_{AMPA}$ augmented the $Cl^-$-influx, while at high $[Cl^-]_i^0$ prolongation of $\tau_{AMPA}$ diminished the $Cl^-$-efflux. Bars represent mean ± SD of 9 repetitions.

## Spatial and temporal constrains of AMPA-mediated infuences on GABAergic $[Cl^-]_i$ transients in a morphologically realistic neuronal model

Next we analyzed how the spatial relation between GABA and AMPA synapse activation influences the size of GDP-dependent $[Cl^-]_i$ transients. For this purpose, we first compared whether direct co-localization of each of the 107 AMPA synapses with a GABA synapse (100% spatial correlation) affects the AMPA-mediated shift in the GDP-dependent $[Cl^-]_i$ transients. These simulations revealed that the activity-dependent $[Cl^-]_i$ shifts with this 100% spatially correlated AMPA synapses were not significantly different from models with a random spatial distribution of AMPA synapses (Fig 9A). Also a restriction of AMPA synapses in either the distal 25% or the proximal 25% of the dendrite length had no substantial effect on the magnitude of activity-dependent $[Cl^-]_i$ transients (S5 Fig). We propose that this lack of an effect was caused by the fact that the overall density of GABA and AMPA synapses in each of the 56 dendrites was so high, that independent of the individual synaptic localization comparable effects on $DF_{GABA}$ were induced. Therefore, we next relocated the GABA and AMPA synapses in distinct parts of the dendritic compartment, with either AMPA synapses located only in the most proximal 12 dendritic branches and the GABA synapses in the most distal 34 dendritic branches, or vice versa (Fig 9B). In these simulations the $GABA_A$ receptor-induced $[Cl^-]_i$ shift without AMPA co-stimulation differ from the previous stimulations, as the $Cl^-$-influx was restricted to a subset of dendrites (Fig 9B). Under this strict regime, the location of the AMPA synapses had a slight effect on $\Delta_G[Cl^-]_i$. With all GABA synapses in the proximal dendrites and the AMPA synapses located in the distal dendrites a co-stimulation induced a $\Delta_G[Cl^-]_i$ of 1.25 ± 0.17 mM (n = 9 repetitions) at a $[Cl^-]_i^0$ of 5 mM (Fig 9B). In contrast, when all GABA synapses were located in the distal dendrites and all AMPA synapses in the proximal dendrites co-stimulation induced a $\Delta_G[Cl^-]_i$ of 0.72 ± 0.07 mM (Fig 9B). A similar tendency was also observed for other $[Cl^-]_i^0$ (Fig 9B). In summary, these results indicate that for a massive stimulation, like the modeled GDP, changes in the spatial correlation between AMPA and GABA receptors have only subtle effects on activity-dependent $[Cl^-]_i$ transients.

Finally, we investigated the effect of the temporal correlation between AMPA and GABA synaptic inputs on the activity-dependent $[Cl^-]_i$ changes. Since it was not possible to generate a non-trivial decorrelation of AMPA and GABA synaptic inputs during GDP-like activity, we stimulated 100 AMPA ($g_{AMPA}$ = 0.305 nS/$\tau_{AMPA}$ = 11 ms) and 100 GABA ($g_{AMPA}$ = 0.789 pS/$\tau_{AMPA}$ = 37 ms) synaptic inputs at a frequency of 20 Hz, using a random temporal distribution

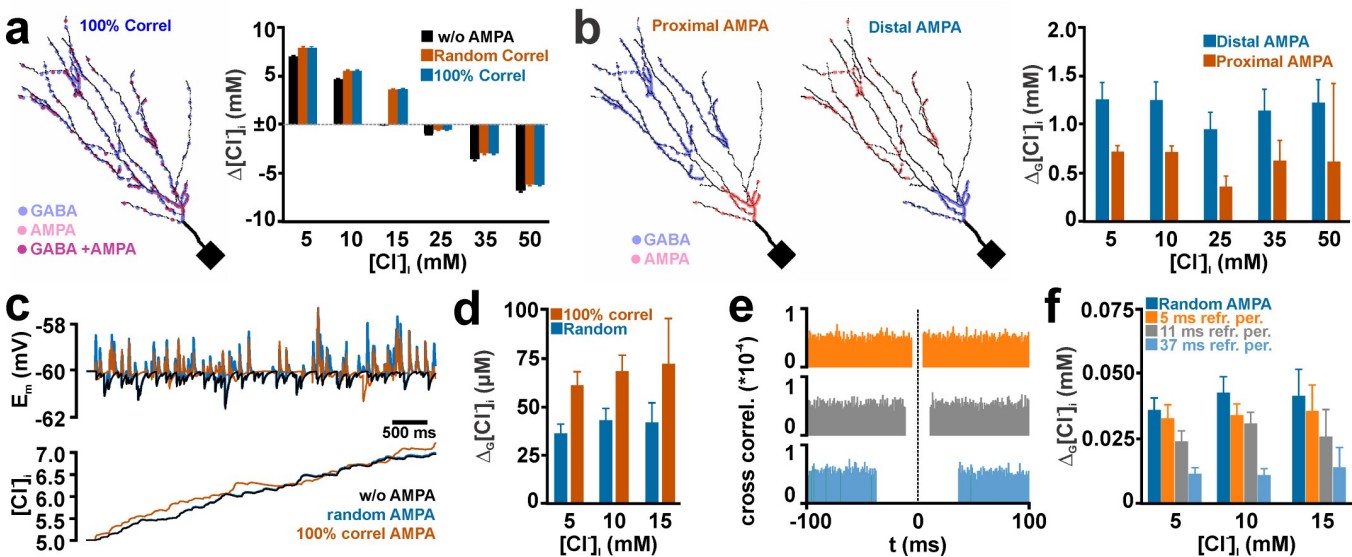

**Fig 9. Effect of spatial and temporal correlation between multiple dendritic GABA and AMPA synapse activation on the activity-dependent $[Cl^-]_i$ transients.** (a) Effect of randomly distributed (orange columns) and GABA colocalized (blue columns) AMPA inputs on GDP-induced $[Cl^-]_i$ transients. Note that both distributions had identical effects. In the left panel the positions of the synapses in the dendritic compartment are plotted. (b) In the left panels the localization of AMPA receptors in proximal and distal dendrites are displayed. Note that GABA synapses were placed in the opposite (distal and proximal) compartment. The right plot indicates that significantly larger effects of AMPA co-stimulation were observed if the AMPA synapses were positioned in the proximal compartment. (c) Typical voltage and $[Cl^-]_i$ traces observed upon random stimulation of GABA synapses without AMPA co-stimulation (black), random AMPA co-stimulation (blue), or with temporally perfectly correlated AMPA inputs (orange). (d) Statistical analysis of the additional $[Cl^-]_i$ increase upon AMPA co-stimulation ($\Delta_G[Cl^-]_i$). (e) Cross correlograms of simulations for refractory period of 5 ms (orange), 11 ms (gray), and 37 ms (blue) between GABA and AMPA stimuli. The relative occurrence of AMPA stimuli in relation to each GABA stimulus was plotted, demonstrating the lack of AMPA receptor mediated synaptic input in the given refractory intervals. (f) Statistical analysis of $\Delta_G[Cl^-]_i$ at different refractory periods. Note that at refractory periods of $\geq 11$ ms significantly smaller $\Delta_G[Cl^-]_i$ occurred. Bars represent mean ± SD of 9 repetitions.

for both types of synapses. While at a $[Cl^-]_i^0$ of 5 mM the random stimulation of only GABA synapses induced a $[Cl^-]_i$ increase by 2.06 ± 0.16 mM (n = 9 repetitions), this increase was augmented by 0.04 ± 0.005 mM upon random co-stimulation of AMPA synapses (Fig 9C and 9D. Comparable $\Delta_G[Cl^-]_i$ values were also observed for $[Cl^-]_i^0$ of 10 mM and 15 mM. For higher $[Cl^-]_i^0$ the trend towards larger $[Cl^-]_i$ at AMPA co-stimulation maintained. However, the large variance in the individual maximal $[Cl^-]_i$ responses in each simulation and the resulting high SD values for $\Delta_G[Cl^-]_i$ values caused that these changes were not significant. If in this stimulation paradigm the AMPA stimuli occurred exactly at the same time points as the GABAergic inputs (i.e. 100% temporal correlation), $\Delta_G[Cl^-]_i$ was significantly increased to 0.065 ± 0.008 mM (at a $[Cl^-]_i^0$ of 5 mM), to 0.078 ± 0.009 mM (at 10 mM $[Cl^-]_i^0$), and to 0.084 ± 0.034 mM (at 15 mM $[Cl^-]_i^0$) (Fig 9D).

To investigate the effect of a decorrelation between GABA and AMPA inputs, we implemented an algorithm that generates refractory periods of 5, 11, and 37 ms around each GABAergic stimulus in which AMPA inputs are omitted (Fig 9E). These experiments revealed that for a $[Cl^-]_i^0$ of 5 mM the $\Delta_G[Cl^-]_i$ values are significantly decreased from 0.041 ± 0.005 mM (n = 9 repetitions at random correlation) to 0.027 ± 0.004 at a refractory period of 11 ms, and to 0.014 ± 0.003 at a refractory period of 37 ms (Fig 9F). At a refractory period of 5 ms $\Delta_G[Cl^-]_i$ decreased only slightly and not significantly to 0.037 ± 0.006. A comparable reduction was also observed for a $[Cl^-]_i^0$ of 10 mM (Fig 9F). At 15 mM only for a refractory period of 37 ms a significant reduction in $\Delta_G[Cl^-]_i$ was observed (from 0.048 ± 0.026 mM to 0.016 ± 0.011 mM). In summary, these results support the finding that the temporal correlation between GABA and AMPA synapses enhance activity-dependent $[Cl^-]_i$ shifts, in particular if AMPA and GABA inputs are correlated within the time range of their decay time constants.

## Discussion

Recent studies provide increasing evidence that GABAergic responses show an activity- and compartment-dependent behavior due to ionic plasticity in $[Cl^-]_i$ [11, 24]. Here we used biophysical modeling to study how coincident glutamatergic inputs enhance the $[Cl^-]_i$ changes induced by GABAergic activation. The main findings of this computational study can be summarized as follows: 1.) Glutamatergic co-stimulation had a direct effect on the GABAergic $[Cl^-]_i$ changes, thereby enhancing $Cl^-$ influx at low $[Cl^-]_i^0$ and attenuating or even reversing the $Cl^-$ efflux caused at high $[Cl^-]_i^0$. 2.) Massive glutamatergic co-stimulation promotes $Cl^-$ influx at all $[Cl^-]_i^0$, whereas physiological levels of glutamatergic co-stimulation mediate biphasic $[Cl^-]_i$ changes at intermediate $[Cl^-]_i$ levels typical for immature neurons. 3.) Keeping the decay kinetics of glutamatergic inputs below that of GABAergic inputs attenuated the $[Cl^-]_i$ changes. 4.) The spatial and temporal correlation between glutamatergic and GABAergic inputs has a substantial influence on the $[Cl^-]_i$ changes, with a surprisingly large temporal interval with correlation-independent $[Cl^-]_i$ changes. 5.) In a morphologically realistic model with physiologically relevant synaptic activity that replicates GDPs, the conductance and kinetics of correlated glutamatergic activity have a substantial impact on the $[Cl^-]_i$ changes. 6.) While the spatial correlation between distributed GABA and glutamatergic synapses has only a minor effect on activity-dependent $[Cl^-]_i$ changes, their temporal correlation has strong effects. 7.) Besides AMPA receptors also NMDA receptors and dendritic action potentials can enhance the GABAergic $[Cl^-]_i$ changes.

In the present model we emulated the membrane transport processes by two exponentially decaying functions [25, 29]. This model offers the advantage that it is computationally less demanding than other approaches, which model the $[Cl^-]_i$ dynamics by a set of transmembrane ion transporters that influence $[Cl^-]_i$ [27, 45]. On the other hand, our model does not consider additional effects that can be mediated by the influence of the glutamatergic $Na^+$ and $K^+$ fluxes on the NKCC1 and KCC2-based $[Cl^-]_i$ homeostasis [46]. While these oppositely directed $Na^+$ and $K^+$ fluxes will lead to balanced effects in the NKCC1-based $[Cl^-]_i$ homeostasis typical for immature neurons, it will attenuate the driving force for $Cl^-$ extrusion in the mature KCC2-based $[Cl^-]_i$ homeostasis. Presumably, larger $[Cl^-]_i$ increases can be expected when more realistic transport processes are considered for the $[Cl^-]_i$ models. In addition, we used a purely diffusional model without considering electrodiffusion [26, 27], as it has been demonstrated recently that for $Cl^-$ ions the diffusional movement is considerably larger than the contribution of electric drift [46].

In this study we investigated the effect of co-activation of glutamatergic AMPA receptors on GABA receptor-dependent $[Cl^-]_i$ changes over a wide range of $[Cl^-]_i^0$, which gave us the opportunity to evaluate the role of ionic plasticity for mature and developing nervous systems [3, 47]. In the mature brain GABAergic activity causes an increase in $[Cl^-]_i$ [11, 16, 25, 48, 49], thereby decreasing the inhibitory action of the GABAergic system [13, 26, 27, 33]. Note in this respect that inhibition, as defined by a decrease in spike probability, is not necessarily related to a hyperpolarization, but that shunting inhibition considerably contributes to inhibition [1, 42, 50]. The present study demonstrates that glutamatergic co-stimulation enhances the GABAergic $Cl^-$-influx and thus enhances the activity-dependent $[Cl^-]_i$ increase, as has recently been shown in-vitro [32] and in-silico [33]. As a consequence, the inhibitory effect of GABA may be even more impaired upon a co-activation of glutamate receptors [33]. The high basal $[Cl^-]_i$ in immature neurons leads to GABAergic $Cl^-$-efflux and thus to a decrease in $[Cl^-]_i$ upon GABAergic activation [18, 51]. This $[Cl^-]_i$ decline can be associated with a loss of excitatory action and/or an enhancement of the shunting inhibitory effect of GABA receptors [21, 50, 52]. The present study demonstrates for the first time that in immature neurons a co-activation

of GABA and glutamate receptors attenuates the activity-dependent $[Cl^-]_i$ decrease, thereby stabilizing the depolarizing actions of GABA.

The absolute values of the additional $[Cl^-]_i$ changes induced by AMPA co-stimulation in our ball-and-stick simulations are small, however it has to be taken into account that these changes represent the $[Cl^-]_i$ changes induced at a single synapse with a single synaptic stimulus. Physiological levels of synaptic activity may thus cause significantly larger $[Cl^-]_i$ changes, as has been shown in-vitro [32]. Even physiological levels of glutamatergic co-stimulation (0.305–3.05 nS, representing 1-10x spontaneous synaptic inputs) mediate in our model a reliable increase in $Cl^-$ influx at low $[Cl^-]_i^0$, implicating that glutamatergic co-stimulation will enhance ionic plasticity in mature neurons. Thereby coincident glutamatergic activity provides an additional challenge for the $[Cl^-]_i$ homeostasis in mature neurons and can substantially contribute to an activity-dependent decline in the inhibitory action of GABAergic synapses [33, 53, 54]. But even small $Cl^-$ changes can cause substantial functional alterations for neuronal computation, as has been recently demonstrated in-silico [45, 55]. Subtle changes in GABAergic membrane responses can directly interfere with the action potential threshold [56], but may also affect the temporal fidelity of inhibitory synaptic inputs, as with a weakening of GABAergic inhibition the temporal window for effective inhibition of excitatory inputs became smaller [52].

In addition, we were also able to demonstrate that several other depolarizing events can augment the GABAergic $[Cl^-]_i$ changes. Co-activation of NMDA receptors augment the $[Cl^-]_i$ changes with a steeper dependency on the membrane conductance as compared to AMPA receptors. This behavior is probably reflecting the $Mg^{2+}$ block of NMDA receptors [40], leading to smaller inward currents as long as $E_m$ was below the threshold for the release of the $Mg^{2+}$ block. Accordingly, at moderate conductances NMDA receptors mediate smaller effects on the $[Cl^-]_i$ changes than AMPA receptors. But under physiological conditions, with AMPA-NMDA receptor co-activation, we assume that NMDA receptors will provide an additional drive for larger $[Cl^-]_i$ changes. Accordingly, we observed in the reconstructed neurons that the simulated GDP-like activity generated larger $[Cl^-]_i$ changes when the AMPA-receptors were replaced with NMDA-receptors, because under this condition the dendritic depolarizations were sufficient to effectively remove the $Mg^{2+}$ block [40]. In addition, we were able to demonstrate that action potential firing, generated by a Hodgkin-Huxley-like mechanism, mediates an additional increase in the $[Cl^-]_i$ changes. This suggests that under physiological conditions burst firing can substantially contribute to ionic plasticity. In contrast, the addition of a tonic GABAergic conductance had only a negligible effect of the activity-dependent $[Cl^-]_i$ changes.

As mentioned before, the high $[Cl^-]_i$ in immature neurons and the resulting GABAergic $Cl^-$-efflux results in a condition in which glutamatergic co-stimulation attenuates GABAergic $[Cl^-]_i$ loss and thereby stabilizes depolarizing GABAergic actions. In addition, our computational model demonstrates that glutamatergic co-stimulation at physiologically relevant levels ($g_{AMPA}$ of 0.305 to 3.05 nS, corresponding to 1 to 10x of experimentally determined conductance for spontaneous AMPA receptor mediated inputs) causes biphasic $[Cl^-]_i$ changes at $[Cl^-]_i$ values between 15 and 35 mM (i.e. in the typical range determined in immature neurons [4, 57, 58]). These biphasic $[Cl^-]_i$ changes rely on the fact that the activation of glutamatergic receptors transiently pushes the membrane potential above $E_{Cl}$ and thus supports a $Cl^-$-influx during this interval. The biphasic $[Cl^-]_i$ changes reduce the maximal $[Cl^-]_i$ decline at the end of the GABAergic postsynaptic potential. By this process the activity-dependent $[Cl^-]_i$ decline as well as the burden of glutamatergic co-stimulation on $[Cl^-]_i$ homeostasis is reduced. It is tempting to speculate that this limitation of activity-dependent $Cl^-$-fluxes may be one reason for the inefficient $Cl^-$-transport in the immature CNS [4, 22]. The functional implications of

activity dependent $[Cl^-]_i$ changes in immature neurons are harder to predict, as depolarizing GABAergic responses can mediate inhibitory [59, 60] as well as excitatory actions [61–63]. However, it can be estimated that an attenuation of the GABAergic $[Cl^-]_i$ depletion will prevent/ameliorate a loss of GABAergic excitation and will prevent/ameliorate an increase in the inhibitory effect of depolarizing GABAergic responses. Thereby coincident glutamatergic activity, which is a typical feature of the correlated network events in developing neuronal systems [36, 37], can stabilize GABAergic functions.

Our simulation demonstrated that $\tau_{AMPA}$ has, as expected, a substantial influence on $\Delta_G[Cl^-]_i$, with faster AMPA events reducing the amount of ionic plasticity. Thus the shorter AMPA receptor-mediated responses at adult synapses [64, 65] can reduce the impact of AMPA/GABA co-activation on $[Cl^-]_i$ homeostasis, in addition to making transmission more precise. In this respect, it is also tempting to speculate that conditions which permit the unblocking of NMDA-receptors (a subtype of glutamate receptors that is characterized by a rather long decay and that plays an essential role in learning, [66]), will also lead to more massive burden on $[Cl^-]_i$ homeostasis. The resulting short-term decline in GABAergic inhibition will make neurons more prone to the induction of long-term potentiation.

In addition to this expected influence of $\tau_{AMPA}$ on ionic plasticity, our simulations revealed a complex dependency of $\Delta_G[Cl^-]_i$ on $\tau_{GABA}$ and $\tau_{AMPA}$. Interestingly the maximal $\Delta_G[Cl^-]_i$ values were obtained when $\tau_{GABA}$ was slightly larger than $\tau_{AMPA}$. An additional prolongation of $\tau_{GABA}$ even reduced $\Delta_G[Cl^-]_i$ and thus diminished ionic plasticity. Intriguingly, such a prolongation of $\tau_{GABA}$ led to the appearance of a "plateau"-like phase, in which $\Delta_G[Cl^-]_i$ is insensitive to the latency between GABA and AMPA inputs. The analysis of the $Cl^-$-fluxes in the phase diagram revealed that this "plateau"-like phase was due to the fact, that the $Cl^-$-fluxes converged over a wide range of AMPA/GABA latencies in the trajectory of $[Cl^-]_i$ changes obtained by synchronous GABAergic and glutamatergic stimulation. For short $\tau_{AMPA}$ no "plateau"-like phase could be observed because under these conditions the time course of $E_m$ was mainly determined by the membrane time constant. This explanation was supported by our observation that an artificial increase in the membrane time constant by an augmented $C_m$ caused a dissipation in the phase plane plot from two attractors towards more dissociated trajectories. For larger $\tau_{AMPA}$ of 37 and 50 ms such a "plateau"-line phase was not reached because at this longer time constant the decay of the voltage deflection is determined by both the inactivation of AMPA- and $GABA_A$ receptors and thus a gradual decrease in $DF_{GABA}$ occurred during the course of co-activation.

It has been suggested that under mature conditions ionic plasticity of $[Cl^-]_i$ can act as a coincidence detector for simultaneous GABAergic and glutamatergic synaptic inputs [32]. Via such a coincidence detection GABAergic inhibition will be particularly attenuated by coincident glutamatergic inputs. The resulting $[Cl^-]_i$ increase and the corresponding reduction in the inhibitory GABAergic capacity will subsequently promote the relay of excitation in neuronal networks. The intriguing finding of the present manuscript, that the effect of glutamatergic co-stimulation on the GABAergic $[Cl^-]_i$ increase was stable for a considerable latency interval between GABAergic and glutamatergic inputs would allow the system to use a stable mechanism for adjusting the gating of excitatory information. In this respect, the striking asymmetry in $\Delta_G[Cl^-]_i$ under physiological conditions of the decay times ($\tau_{AMPA} < \tau_{GABA}$) implies a spike-time dependency of ionic plasticity. While glutamatergic inputs preceding GABAergic inputs have only a small effect on $\Delta_G[Cl^-]_i$, a stable shift in $\Delta_G[Cl^-]_i$ was induced when glutamatergic synapses were activated during the decay phase of GABAergic responses. A possible functional implication of this spike-time dependency would be that common feed-forward inhibitory circuits, in which synaptic inhibition always occurs after the glutamatergic inputs, will be only minimally affected by ionic plasticity, guaranteeing an efficient and stable feed-

forward inhibition. Only in cases in which glutamatergic excitation was induced during ongoing GABAergic stimulation, a substantial ionic plasticity would operate. By this mechanism the inhibitory capacity of GABA will be reduced particularly in the postsynaptic neurons that receive frequent glutamatergic inputs, thereby facilitating particularly these pathways.

In summary, our results demonstrate that glutamatergic co-activation has a prominent time- and space-dependent effect on the amount of GABA-receptor mediated $[Cl^-]_i$ changes. This ionic plasticity depends on the properties of glutamatergic inputs and their temporal correlation with the GABAergic inputs. These glutamatergic modulations of GABAergic ionic plasticity could possibly contribute to short-term memory and are likely to influence information processing in the developing and mature nervous system.

## Materials and methods

### Compartmental modeling

The biophysics-based compartmental modeling was performed using the NEURON environment (neuron.yale.edu). The source code of models and stimulation files used in the present paper can be found in ModelDB (http://modeldb.yale.edu/266823). For compartmental modelling we used in the first simulations a simple ball-and-stick model (soma with d = 20 μm, linear dendrite with l = 200 μm, diameter 1 μm, and 103 nodes; *cell_soma_dendrite. hoc*). In the experiment investigating the role of spatial orientation between GABA and AMPA synapses, the length of the dendrite was increased to 1000 μm (*cell_soma_dendrite_long.hoc*). In further experiments we used a reconstructed CA3 pyramidal cell (*Cell1_Cl-HCO3_Pas.hoc*; [17]). We used the reconstruction of an immature CA3 pyramidal neuron for this purpose, because coincident GABAergic and glutamatergic activity used for modelling of ionic plasticity was recorded in exactly this neuron population [17]. This reconstructed neuron contained a soma (d = 15 μm), a dendritic trunk (d = 2 μm, l = 32 μm, 9 segments) and 56 dendrites (d = 0.36 μm, 9 segments each). In all of these compartments a specific axial resistance ($R_a$) of 34.5 Ωcm and a specific membrane capacitance ($C_m$) of 1 μFcm$^{-2}$ were implemented. $C_m$ was varied in two experiments to accelerate/decelerate the membrane time constant. A specific membrane conductance ($g_{pas}$) of 1.0 mS/cm$^2$ with a reversal potential of -60 mV was inserted in all neuronal elements. In some experiments we additionally implemented a tonic background conductance of 8.75 μS/cm$^2$ [43] and reduced the passive conductance to 0.99125 mS/cm$^2$, to maintain the input resistance of 188.2 MΩ.

GABA$_A$ synapses were simulated as a postsynaptic parallel Cl$^-$ and HCO$_3^-$ conductance with exponential rise and exponential decay:

$$I_{GABA} = I_{Cl} + I_{HCO3} = 1/(1 + P) \cdot g_{GABA} \cdot (V - E_{Cl}) + P/(1 + P) \cdot g_{GABA} \cdot (V - E_{HCO3})$$

where P is a fractional ionic conductance that was used to split the GABA$_A$ conductance ($g_{GABA}$) into Cl$^-$ and HCO$_3^-$ conductance. $E_{Cl}$ and $E_{HCO3}$ were calculated from the Nernst equation. The GABA$_A$ conductance was modeled using a two-term exponential function, using separate values of rise time (0.5 ms) and decay time (variable, mostly 37 ms). Parameters used in our simulations were as follows: $[Cl^-]_o$ = 133.5 mM, $[HCO3^-]_i$ = 14.1 mM, $[HCO3^-]_o$ = 24 mM, temperature = 31˚C, P = 0.18. The conductance values for GABA synapses varied as given in the main text. AMPA synapses were modeled by an Exp2Syn point process using a reversal potential of 0 mV, a tau 1 value of 0.1 ms and a tau2 value of 11 ms, in accordance with the experimentally determined value [17], except where noted. The conductance values for AMPA synapses varied as given in the main text.

For the ball-and-stick model a single GABA$_A$ synapse was placed in the middle of the dendrite, except where noted. The AMPA synapse was placed as indicated in the figure legends.

For the simulation of a GDP in the reconstructed CA3 neuron 534 GABAergic synapses and 107 AMPA synapses were randomly distributed within the dendrites of the reconstructed neuron, except where noted. The number of GABA and AMPA synapses replicate the estimated synapse numbers during experimentally recorded GDPs [17]. The properties of these synapses were given in the results part and/or the corresponding figure legends. Nine repetitions of these simulations were generated by altering the seed values for the random functions, which resulted in a redistribution of all synapses within the dendritic compartment and the timing of their activation. The results of these repeated simulations were given and displayed as mean ± SD.

For each synapse, the index of the dendrite as well as the position within this dendrite was randomly determined using a normal distribution. The time points of GABA and AMPA inputs were determined stochastically using a normal distribution ($\mu$ = 600 ms, $\sigma$ = 9000 ms for GABA and $\mu$ = 650 ms, $\sigma$ = 8500 ms for AMPA), which emulates the distribution of GABAergic and glutamatergic PSCs during a GDP observed in immature hippocampal CA3 pyramidal neurons [17]. In a few simulated experiments, the 107 AMPA inputs were stimulated synchronously with a random subgroup of the GABA inputs. Since it was not possible to generate a non-trivial decorrelation of AMPA and GABA synaptic inputs during GDP-like activity, for the decorrelation experiments, we stimulated 100 AMPA and 100 GABA synaptic inputs at a frequency of 20 Hz, using a random temporal distribution for both synapses. For the decorrelation, we used a simple algorithm that generated new random numbers until the time point of an AMPA input was temporally separated by more than the given refractory periods with respect to the time points of all GABA inputs.

For analyzing the effect of NMDA receptors, we included an established model for the NMDA receptor (Exp2NMDA2 from Senselab Model DB ID: 145836 [39]) with an onset time constant of 8.8 ms and a $\tau_{NMDA}$ of 500 ms, using similar conductance values for $g_{NMDA}$ as for the AMPA receptors. In addition, in a limited set of simulated experiments we inserted an established Hodgkin-Huxley based model of hippocampal action potentials (Model DB ID: 3263 [41]) into soma and dendrite of the ball-and-stick model to emulate somatic and back-propagating dendritic action potentials. This Hodgkin-Huxley-based model also includes a voltage-gated $Ca^{2+}$ conductance, which we also implemented separately in soma and dendrite of the ball-and-stick model to investigate the contribution of $Ca^{2+}$ conductances to $[Cl^-]_i$ transients.

For the modeling of the $GABA_A$ receptor-induced $[Cl^-]_i$ and $[HCO_3^-]_i$ changes, we calculated ion diffusion and uptake by standard compartmental diffusion modeling. In order to keep the computational complexity at reasonable levels, we used a purely diffusional model without considering electrodiffusion [26, 27], as recently it has been demonstrated that for $Cl^-$ ions the diffusional movement is considerably larger than the contribution of electric drift [46]. To simulate intracellular $Cl^-$ and $HCO_3^-$ dynamics, we adapted our previously published model [25, 30]. Longitudinal $Cl^-$ diffusion along dendrites was modeled as the exchange of anions between adjacent compartments. For radial diffusion, the volume was discretized into a series of four concentric shells around a cylindrical core and $Cl^-$ or $HCO_3^-$ was allowed to flow between adjacent shells. The free diffusion coefficient for $Cl^-$ inside neurons was set to 2 $\mu m^2$/ms [14] and for $HCO_3^-$ to 1.18 $\mu m^2$/ms [67]. To simulate transmembrane transport of $Cl^-$ and $HCO_3^-$, we implemented an exponential relaxation process for $[Cl^-]_i$ and $[HCO_3^-]_i$ to resting levels $[Cl^-]_i^{rest}$ or $[HCO_3^-]_i^{rest}$ with a time constant $\tau_{Ion}$.

$$\frac{d[Ion^-]_i}{dt} = \frac{[Ion^-]_i^{rest} - [Ion^-]_i}{\tau_{Ion}}$$

$Cl^-$ transport was in most experiments (if not otherwise noted) modeled as bimodal process, for $[Cl^-]_i < [Cl^-]_i^{rest}$ $\tau_{Ion}$ was set to 174 s to emulate an NKCC1-like $Cl^-$ transport mechanism. For $[Cl^-]_i > [Cl^-]_i^{rest}$ $\tau_{Ion}$ was set to 321 s to emulate passive $Cl^-$ efflux [43].

The impact of GABAergic $Cl^-$ currents on $[Cl^-]_i$ and $[HCO_3^-]_i$ was calculated as:

$$\frac{d[Ion^-]_i}{dt} = \frac{1}{F}\frac{I_{Ion}}{volume}$$

To simulate the GABAergic activity during a GDP, a unitary peak conductance of 0.789 nS and a decay of 37 ms were applied to each GABAergic synapse, in accordance with properties of spontaneous GABAergic postsynaptic currents in CA3 pyramidal neurons [17].

For the analysis and display of the data in the ball-and-stick model we used $E_m$ and $[Cl^-]_i$ values at the site of the GABAergic synapses. For the reconstructed CA3 neuron we used the somatic $E_m$ and the mean $[Cl^-]_i$ within the dendritic compartment. The mean $[Cl^-]_i$ of all dendrites was calculated by averaging the $[Cl^-]_i$ at 50% of dendritic length for all 56 dendrites. This procedure mimics the experimental procedure of Lombardi et al. [17], who determined $E_{GABA}$ by focal application in the dendritic compartment.

For the calculation of $\Delta[Cl^-]_i$, the maximal deviation of $[Cl^-]_i$ upon a GABAergic stimulus ($[Cl^-]_i^S$) was subtracted from the resting $[Cl^-]_i$ before the stimulus ($[Cl^-]_i^R$). For biphasic responses both minimal and maximal $[Cl^-]_i^R$ were determined and displayed. In some cases, the manifest $\Delta[Cl^-]_i$ for these responses was calculated as:

$$\Delta[Cl^-]_i = [Cl^-]_i^{S,min} - [Cl^-]_i^R \; if \; \text{abs}([Cl^-]_i^{S,min}) > \text{abs}([Cl^-]_i^{S,max})$$

$$\Delta[Cl^-]_i = [Cl^-]_i^{S,max} - [Cl^-]_i^R \; if \; \text{abs}([Cl^-]_i^{S,min}) \leq \text{abs}([Cl^-]_i^{S,max})$$

To quantify the influence of AMPA co-stimulation on the GABAergic $[Cl^-]_i$ transients, the amount of additional $[Cl^-]_i$ changes induced by AMPA co-stimulation ($\Delta_G[Cl^-]_i$) was calculated from the $\Delta[Cl^-]_i$ values with/without AMPA co-stimulation as follows:

$$\Delta_G[Cl^-]_i = \Delta[Cl^-]_i^{AMPA/GABA} - \Delta[Cl^-]_i^{GABA}$$

## Supporting information

**S1 Fig. Effect of the membrane time constant on the trajectories of $[Cl^-]_i$ transients in a phase-plane plot.** The black lines represent the $[Cl^-]_i$ changes induced by stimulation of GABA synapses only. Typical voltage deflections are displayed in the insets. Note that decreasing the membrane time constant (panel a) resulted in a comparable convergence of the trajectories towards 0 ms latency (purple line) and GABA only conditions (black lines) as under control conditions (panel b), despite the sharper trajectories. In contrast, after prolonging the membrane time constant (panel c) the trajectories did not converge to the 0 ms latency condition at the intersection with $\partial[Cl^-]_i /\partial t = 0$ mM/s (dashed line).
(TIF)

**S2 Fig. Effect of AMPA receptor activation on $[Cl^-]_i$ in a ball-and-stick model with a tonic GABAergic background conductance of 8.75 nS/cm².** Conductance and time constant of the AMPA receptor-mediated inputs were systematically varied as indicated in the diagram. Note the minimal $[Cl^-]_i$ changes in the µM range under these conditions. At physiological values for AMPA inputs ($g_{AMPA} = 0.305$ nS, $\tau_{AMPA} = 11$ ms) a $[Cl^-]_i$ change of 0.1 µM was induced.
(TIF)

**S3 Fig. Effect of AMPA receptor co-stimulation on GABA$_A$ receptor-induced [Cl$^-$]$_i$ transients in a morphologically realistic neuronal model with a tonic GABAergic background conductance of 8.75 nS/cm$^2$.** $g_{AMPA}$ and [Cl$^-$]$_i^0$ was varied as indicated in the graph, all other values were kept constant at physiological values ($g_{GABA}$ = 0.789 nS, $\tau_{GABA}$ = 37 ms, $\tau_{AMPA}$ = 11 ms). Addition of the tonic GABA conductance had no obvious effect on the activity-dependent [Cl$^-$]$_i$ transients.
(TIF)

**S4 Fig. Effect of NMDA receptor co-activation on GABA$_A$ receptor-induced [Cl$^-$]$_i$ transients in a morphologically realistic neuronal model.** (a) Typical voltage deflections during a GDP at an [Cl$^-$]$_i^0$ of 5 mM (left panel) and 25 mM (right panel) using different strength of NMDA co-stimulation as indicated by the color code. Note the substantial shift towards depolarized potentials at high $g_{NMDA}$. (b) Time course of average dendritic [Cl$^-$]$_i$ at different [Cl$^-$]$_i^0$ and $g_{NMDA}$. Note that at both [Cl$^-$]$_i^0$ the maximal [Cl$^-$]$_i$ change was augmented by addition of 107 NMDA synapses. (c) Statistical analysis of the voltage changes induced by simulated GDPs with different [Cl$^-$]$_i^0$ and $g_{NMDA}$. (d) Statistical analysis of [Cl$^-$]$_i$ changes induced by simulated GDPs with the given [Cl$^-$]$_i^0$ and $g_{NMDA}$ (closed bars) as compared to the [Cl$^-$]$_i$ changes with AMPA receptor co-stimulation (open bars). Bars represent mean ± SD of 9 repetitions.
(TIF)

**S5 Fig. Effect of AMPA receptor co-stimulation on GABA$_A$ receptor-induced [Cl$^-$]$_i$ transients in a morphologically realistic neuronal model, in which the 107 AMPA synapses were positioned either in the distal or proximal quarter of each dendrite and the 534 GABA synapses in the remaining 75% of dendritic length to implement partial spatial decorrelation.** Parameters are set to physiological values ($g_{GABA}$ = 0.789 nS, $\tau_{GABA}$ = 37 ms, $g_{AMPA}$ = 0.305 nS, $\tau_{AMPA}$ = 11 ms). (a) illustrates that the GABA$_A$ receptor-induced [Cl$^-$]$_i$ transients are slightly altered with the reposition of the GABA synaptic sites. (b) The increase of [Cl$^-$]$_i$ transients by AMPA co-stimulation was virtually unaffected by this mild spatial decorrelation.
(TIF)

## Acknowledgments

The authors thank Beate Krumm for her excellent technical support.

## Author Contributions

**Conceptualization:** Peter Jedlicka, Werner Kilb.

**Formal analysis:** Aniello Lombardi, Peter Jedlicka, Werner Kilb.

**Writing – original draft:** Aniello Lombardi, Peter Jedlicka, Werner Kilb.

**Writing – review & editing:** Peter Jedlicka, Heiko J. Luhmann, Werner Kilb.

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
