## [Decision Letter · Decision Letter 0]

21 Sep 2020

Dear Dr. Kilb,

Thank you very much for submitting your manuscript "Coincident glutamatergic depolarizations enhance GABAA receptor-dependent Cl- influx in mature and suppress Cl- efflux in immature neurons" for consideration at PLOS Computational Biology. As with all papers reviewed by the journal, your manuscript was reviewed by members of the editorial board and by several independent reviewers. The reviewers appreciated the attention to an important topic. Based on the reviews, we are likely to accept this manuscript for publication, providing that you modify the manuscript according to the review recommendations.

Also thanks for taking part in the reproducibility pilot, and congratulations for passing the reproducibility test (this is not always the case).

Sincerely,

Hugues Berry

Associate Editor

PLOS Computational Biology

Daniele Marinazzo

Deputy Editor

PLOS Computational Biology

[LINK]

Reviewer's Responses to Questions

**Comments to the Authors:**

Reviewer #1: The Reproducibility report has been uploaded as an attachment.

Reviewer #2: Review of manuscript: “Coincident glutamatergic depolarizations enhance GABAA receptor-dependent Cl- influx in mature and suppress Cl- efflux in immature neurons”

The authors perform a systematic exploration of how GABA and Glutamate activations at different time points and locations affect the intracellular Cl- concentration in a hippocampal neuron. This is relevant because the GABA-current depends on the transmembrane concentration gradient of Cl-, and changes in intracellular Cl- leads to a so-called “ionic (synaptic) plasticity” of GABA synapses.

The results section consists of a large number of simulations where different parameters (locations, timing, synaptic conductances, synaptic time constants, the membrane time constant, initial Cl- concentration) are varied, mostly one (or two) at the time. It is not easy to boil all that down to any general take-home message, since the effects will always depend on this or that, but I suppose that a general take-home message might not be expected for an intricate system like this.

The main finding seems solid, though, and is reflected in the manuscript title. When the initial Cl- concentration is low/intermediate (mature neurons), Glutamatergic depolarizations tend to enhance Cl- influx during GABAergic activation. This causes an “ionc plasticity” that decreases the inhibitory action of GABA. Differently, when the initial Cl- concentration is high (immature neurons), GABA activation has a depolarizing effect. Glutamatergic depolarizations then tend to suppress Cl- efflux during GABA-activation, thus preserving the ion reservoir (battery), and stabilizing the deploarizing role of GABA.

The modeling work seems solid, and the paper is well written. I think it is suited for publication in PLoS CB, but I have a some (relatively minor) suggestions to how it might be improved:

1: I get the feeling that the paper could be shortened and written in a way that puts more focus around the key findings. Some of the systematic parameter explorations are nice to include, but do not always lead to all that interesting results. Parts of it might be hidden away as supplementary material and simply summarized in the body text. I do not regard this as a criterion for publication though – just a suggestion that the authors could consider.

2: As Fig. 4 (not surprisingly) shows, the Glutamate-effect on Cl- during AMPA activation is solely due to the membrane depolarization that it evokes. The question is then: Is the EPSP from AMPA-activation generally the main source of membrane depolarization in the dendrites of CA3-neurons? What about back-propagating action potentials, dendritic Ca2+ spikes, or (local) secondary effect of AMPA activation mediated by locally present ion channels? Would these phenomena play an equally important role for ionic plasticity?

I think this issue should be addressed, at least as a passage in the discussion. Perhaps it is

beyond the scope of the current paper to actually explore such effects, but I guess it would be possible to do so, using some existing CA3-neuron model with active dendrites, such as the one by Traub et al. 1991 [Journal of Neurophysiology. 1991; 66(2):635–650], Pinsky & Rinzel 1994 [Journal of computational neuroscience. 1994; 1(1-2):39–60], Migliore et al. 1995 [Journal of neurophysiology, 73(3), 1157-1168.], Hemond et al. 2008 [Hippocampus, 18(4), 411-424], or Sætra et al. 2020 [PLoS Comput Biol 16(4): e1007661], the latter of which also includes ion concentration dynamics.

3. [Ca-]i is used in two different meanings - it denotes both the variable intracellular concentration (e.g., on the y-axis in Fig. 1C) and the (constant) initial condition (e.g. in the legend of Fig. 1b). You should make a distinction, and perhaps add an index “0” when the latter is meant.

4. Line 134 refers to the “passive [Ca-]i “. It sounds odd to call a concentration passive. Perhaps “baseline” is better?

5. Line 157 reads: “These simulations revealed that without co-stimulation GABAA receptor induced Cl- fluxes depended only on the initial [Cl-]i (Fig. 2a).” I don’t think anything is revealed here. The initial [Cl-]i is the only thing being varied, so it could not have depended on anything else?

6. Line 344: When you write “speeding up the kinetics of AMPA and GABA-receptor dependent voltage responses”, you make me think that you change the AMPA and GABA time constants, but here I think you have changed the membrane time constant, which is a more general “speeding up of things”. I think you should rephrase this to avoid confusion.

7. Line 364: You refer to your “full” model as “morphologically and biophysically realistic”, although it is a passive model with no ion channels, so it is unclear what kind of biophysical realism you make claim to. How is it more biophysically realistic than e.g., the ball-and-stick model – they differ solely in terms of morphology?

8. Line 636: “the dendrite was electrically and diffusionally detached from the soma”. This means that the model, in practice, did not have any soma, so you did not use a ball-and-stick model, but a stick model. If you only modelled the dendrite, it should be made clear also in the Results-section. Also, I am a bit surprised by this choice. Why cut off the soma? Wouldn´t you expect more realistic results with it being present?

9. In dilute solutions, the diffusion constants for Cl- and HCO3 are 2.03 μm2/ms and 1.18 μm2/ms, respectively (Table 2.1 in Grodzinsky F. Fields, Forces, and Flows in Biological Systems. Garland Science, Taylor & Francis Group, London & New York.; 2011). They could be smaller than that in the cytoplasm (due to obstacles), but I do not think it is possible that they become any larger. Hence, using a value of 2 μm2/ms for HCO3 (Line 675) is probably incorrect. This is not likely to have any severe impact on the simulation results, I imagine, but if it is not too much work, you might want to fix it.

Reviewer #3: This paper uses computational methods to explore how AMPA mediated glutamatergic input affects GABAAR mediated Cl- dynamics in neurons. The paper is well written, clear and logically developed. Although the results perhaps aren’t “earth shattering” or hugely unexpected over what one would predict conceptually, I found the work to be broadly satisfying and certainly publication worthy in this journal;. I found the modelling of GDPs in morphologically realistic immature CA3 neurons particularly good. I also enjoyed the attention payed to how glutamatergic conductances affect neurons with both low and high initial Cl- concentrations. Compared to some other computational models of Cl- dynamics the authors used a rather simple mechanism to describe Cl- extrusion/ intrusion by KCC2/NKCC1, this is not necessarily a problem and I can see how this would have aided the ease of computation in the more morphologically realistic neurons, but I think the authors do need to be a bit more upfront about some of the limitations of the model.

Major comments:

1) In the author’s model, they use a passive conductance (gpas) that does not include Cl- flux. This is despite the fact that tonic Cl- conductances (including tonic GABAAR) exist in neurons. As a result the authors by definition omit a relevant possible route whereby glutamatergic conductances can change [Cl-]I via affecting the driving force for Cl- flux across GABAARs. For example, in section 2.2 which investigates the influence of τAMPA on GABAA receptor induced [Cl-]i transients AMPA conductances with taus longer than the GABAA taus have no effect on the change in [Cl-]I this makes sense in their model as once the GABAAR conductance is over the AMPA driven change on the GABAAR driving force can no longer affect Cl- flux as there is nor GABAAR conductance. In most neurons though tonic GABAAR exists and hence the longer AMPA conductances should progressively change [Cl-]i (not as much as during the phasic GABAAR conductance, but still a bit!). The authors need to address this issue and either model tonic GABAAR, or justify why they did not. Ie Either they need to explicitly ref the fact that tonic GABAAR is small / minimal in immature CA3 neurons hence why they excluded tonic GABAAR conductances and acknowledge that tonic GABAAR in other cell types could affect their findings / make some predictions – ie what would happen in cells with high tonic GABA? Or they need to do some simulations where tonic Cl- flux is also modelled to explore how this variable affects glut-mediated Cl- changes via GABAARs. Ie what is the difference between cells with low tonic GABAA vs high tonic GABAA - for example some version of the simulations in section 2.2, but where tonic Cl- flux is present (there are various ways this could be modelled.) New simulations are not essential but I think attention to this would certainly strengthen the paper.

2) The second limitation of the model which the authors do not state explicitly is the fact that NMDA receptors are not modelled, just AMPA receptors. The authors need to either state why this was the case, or acknowledge this as a limitation? For example how does NMDA change the ball-and-stick model's results. I realise that τAMPA was changed up to 100 ms. But NMDA does have different kinetics, including reliance on Vm.

3) In addition active voltage-gated conductances also weren’t modelled. In the discussion a sentence should be added about how these are likely to add further complexity / another mechanism which could enhance the ability of glutamatergic input to depolarize the neuron and exacerbate Cl- influx via GABAARs.

Minor comments:

I think an important thing to think about is the colours used in the images, which may be difficult to interpret, especially for those with colour blindness. Here are some colour palettes (from seaborn) and their performance with different vision deficiencies: https://gist.github.com/mwaskom/b35f6ebc2d4b340b4f64a4e28e778486

I thought the final sentence of the discussion was too strong and not justified by the papers findings.

These glutamatergic modulations of GABAergic ionic plasticity can contribute to short-term memory and may profoundly influence information processing in the developing and mature nervous system.

Change to

These glutamatergic modulations of GABAergic ionic plasticity could possibly contribute to short-term memory and are likely to influence information processing in the developing and mature nervous system.

Typos:

Line 358: gAPMA -should be gAMPA

Line 381 (i.e. the experimentally determined number and conductance of GABAergic inputs) - should be AMPA inputs

Line 603: inhibition at will – remove “at”

Reviewer #4: I congratulate the authors for their manuscript.

The authors invetigate how coincident excitatory input shapes inhibition and more precisely how Cl- flux associated with gaba currents and subsequent change in chloride concentration is modulated by concurrent excitation. This phenomenon is explained by the fact that excitatory currents depolarize the membrane, which increases the Cl driving force and shapes Cl flux.

The most impressive part of the manuscript is the systematic manner in which the authors investigate the effect of synaptic events time constant, location and synchronisation on Cl- fluxes. They performed a large number of simulations and made extensive anaysis. In particular to my knowledge, the impact of Ampa time constant on Cl- accumulation has not been investigated before.

The topic is important as the results shed light on the determinants of inhibitions and may help better understand how to restore impaired inhibition in pathologies.

That said, the redaction and presentation could be improved at some places. I give here a list specific comments:

1)In the abstract, the author mention that they model immature CA3 neurons. As chloride homeostasis is strongly influenced by maturity, it would be nice if the authors motivated their choice to model an immature neuron.

2) At page 2, lines 43-44, I don't understand the mean of 'destabilization' in the following: 'enhancing the

destabilization of GABAergic inhibition in the mature nervous systems'

3) At the beginning of introduction, line 63. The expression 'Information processing between neurons' is strange. A single neuron processes information but information is transfered between neurons through synapses. There is a disctinction between processing and transfering information.

4) Introduction line 70

'with a high Cl- permeability and a partial HCO3 permeability'.

It is strange to contrast 'high' with 'partial' instead of 'high' with 'low' or 'full' with 'partial'

5) Line 71, what does 'gaba actions' mean? This is a strange wording. Can actions be replaced by a more specific term

6) Line 75, 'the Na+-dependent K+-2Cl--Symporter NKCC1...' it was my belief that the stochiometry of NKCC1 is rather Na+-K+-2Cl-

7) Page 6, lines 123-124. 'leads' and 'allowed' should have the same verb tense.

8) Line 131, DFGABA should be formally defined.

9) Figure 1 a. Here and elswhere, what is shown in the schematic next to GABA and AMPA? is it the time course of current, of conductance or something less? It would be nice to indicate it on the schematic. Scale bars could be added.

10) Figure 2 a and d. These panels are somewhat difficult to understand quickly because of the broken lines corresponding to biphasic responses. The double blue curves in a could look at first glance like a mistake. Maybe there is another way to display the results? Maybe making two different panels for Cl influx and efflux.

11) Comments 9 and 10 are also valid for figure 3.

12) Page 12, lines 235-237. Where is Em recorded? Is it always somatic Em? This could be specified.

13) Line 240-241, the definition of Delta_G[Cl-]i is difficult to read as an inline equation. Maybe it could be defined in text with a full sentence.

14) Page 13, lines 256-257. Is the decrease really exponential? If so, could you explain why? Did you try to fit an exponential and find the constant of the exponential and goodness of fit?

15) Line 263. Why did you choose the range -49ms-100ms?

16) Figure 5 a and b lower parts. I dont like when several curves are superimposed as it provides little information. Could you provide the difference between the curves so we can see how much they are really alike?

17) Page 16 line 320. What exactly do you mean by complex influence? Could you be more specific?

18) Page 17, line 340. What does voltage 'deflections' mean? Could you use a more specific word.

19)I find figure 7 impressive!!

20) In the legend of figure 7, a line break occurs in the middle of [Cl-]i, this should be corrected.

21) When plus-minus values are given, are they standard deviations? It is not clear what is the random part of the simulation. Is the position of the synapses random? Is the timing of synaptic events random? this could be better explained

22) Page 22, lines 455-456

'Therefore, we next located the GABA and AMPA synapses in different parts of the dendritic compartment.'

What does it exactly means? On the same dendritic branch but at different distances from the soma or on different dendritic branches?

23) Figure 9a, bar graph. Is there any difference between the red and blue bars or are they exactly the same? If they are not exactly the same, the authors should find a way to illustrate the difference. From what I understand, each black bar is repeated twice with pairs of black bars having the exact same value. This should be avoided.

24) Fig 9e. It is extremely hard to decipher this panel.

25) Page 23. Line 485-487. The authors should explain in more details what is random in their simulation and eventually what random distribution is used and why.

26) Page 26 line 523. I don't understand what 'stringent' means in this context. Maybe use a more technical and specific word.

27) Line 533, when the authors mention inhibition, do they mean hyperpolarization or decrease in spike rate? This could be specified.

28) Page 26 line 555. The authors mention coding. The notion of coding in this context is quite complex and subtle. The notion of coding could be exposed and explained a bit.

29) Page 27, line 561. The meaning of 'moderate' is not clear for me.

30) Methods line 646. Why not use GHK flux equations which is more accurate when ionic concentrations fluctuate.

31)Page 31, line 663. What exactly follows a Gaussian distribution? Is it the time separating two consecutive onsets of synaptic events?

32) Page 32. Line 690. The long superscript is hard to read. Maybe the equation could be reformated.

33) Page 33, Lines 697-698. 'abs' should not be in italic.

34) Is the formal definition of DFGABA given somewhere in the manuscript?

35) In the abbreviations, the PHCO3 is the relative permeability. Relative to what? This should be written explicitly

**Have all data underlying the figures and results presented in the manuscript been provided?**

Reviewer #1: None

Reviewer #2: Yes

Reviewer #3: Yes

Reviewer #4: Yes

PLOS authors have the option to publish the peer review history of their article (what does this mean?). If published, this will include your full peer review and any attached files.

Reviewer #1: **Yes: **Anand K. Rampadarath

Reviewer #2: **Yes: **Geir Halnes

Reviewer #3: No

Reviewer #4: **Yes: **Nicolas Doyon
---

## [Decision Letter · Decision Letter 1]

30 Nov 2020

Dear Dr. Kilb,

We are pleased to inform you that your manuscript 'Coincident glutamatergic depolarizations enhance GABAA receptor-dependent Cl- influx in mature and suppress Cl- efflux in immature neurons' has been provisionally accepted for publication in PLOS Computational Biology.

Best regards,

Hugues Berry

Associate Editor

PLOS Computational Biology

Daniele Marinazzo

Deputy Editor

PLOS Computational Biology

Reviewer's Responses to Questions

**Comments to the Authors:**

Reviewer #1: Reproducibility report has been uploaded as an attachment.

Reviewer #2: The authors have responded well to my concerns. Congratulations on a nice paper!

Reviewer #3: I am satisfied with the extensive changes and additions performed by the authors in response to my major comments and am therefore in support of publication.

Reviewer #4: A new reading of this manuscript made me appreciate it even more. The technical aspect of the simulations is of great quality. The sheer amount of results and analysis is impressive. The authors have answered satisfactorily toall my comments.

Here are a few minor suggestions

1) Line 82: I think the word 'role' is missing 'Ionic plasticity plays an important for'

2) Line 89: affects should end without an 's'. 'the kinetics of GABAergic responses and the stability

89 of bicarbonate gradients affects the magnitude and'

3) In line 99, affects should also end without an s.

4) Line 208, a space is missing between the seccond 37 and ms. 'sensitive to τAMPA at values < 37 ms, while at values > 37ms'

5) Lines 678-670 not sure of the meaning of 'prevent/ameliorate'.

6) Line 799 maybe the word 'the' is missing between into and and soma 'implemented separately into soma and dendrite'

7) Equations 814, is e[Ion] defined in the text?

8) Line 816-817 the line break should not occur in the middle of [Cl-]i

9) Lines 833, 834, punctuation is missing after equations

**Have all data underlying the figures and results presented in the manuscript been provided?**

Reviewer #1: None

Reviewer #2: Yes

Reviewer #3: None

Reviewer #4: Yes

PLOS authors have the option to publish the peer review history of their article (what does this mean?). If published, this will include your full peer review and any attached files.

Reviewer #1: No

Reviewer #2: **Yes: **Geir Halnes

Reviewer #3: **Yes: **Joseph Raimondo

Reviewer #4: **Yes: **Nicolas Doyon

---

## [Editor Report · Acceptance letter]

8 Jan 2021

PCOMPBIOL-D-20-01408R1 

Coincident glutamatergic depolarizations enhance GABAA receptor-dependent Cl- influx in mature and suppress Cl- efflux in immature neurons

Dear Dr Kilb,

I am pleased to inform you that your manuscript has been formally accepted for publication in PLOS Computational Biology. Your manuscript is now with our production department and you will be notified of the publication date in due course.

With kind regards,

Jutka Oroszlan
